# GRANDE: GRADIENT-BASED DECISION TREE ENSEMBLES FOR TABULAR DATA

**Sascha Marton**
University of Mannheim, Germany
sascha.marton@uni-mannheim.de

**Stefan Lüdtke**
University of Rostock, Germany
stefan.luedtke@uni-rostock.de

**Christian Bartelt**
University of Mannheim, Germany
christian.bartelt@uni-mannheim.de

**Heiner Stuckenschmidt**
University of Mannheim, Germany
heiner.stuckenschmidt@uni-mannheim.de

## ABSTRACT

Despite the success of deep learning for text and image data, tree-based ensemble models are still state-of-the-art for machine learning with heterogeneous tabular data. However, there is a significant need for tabular-specific gradient-based methods due to their high flexibility. In this paper, we propose GRANDE, **GRA**dieN**t**-Based **D**ecision Tree **E**nsembles, a novel approach for learning hard, axis-aligned decision tree ensembles using end-to-end gradient descent. GRANDE is based on a dense representation of tree ensembles, which affords to use backpropagation with a straight-through operator to jointly optimize all model parameters. Our method combines axis-aligned splits, which is a useful inductive bias for tabular data, with the flexibility of gradient-based optimization. Furthermore, we introduce an advanced instance-wise weighting that facilitates learning representations for both, simple and complex relations, within a single model. We conducted an extensive evaluation on a predefined benchmark with 19 classification datasets and demonstrate that our method outperforms existing gradient-boosting and deep learning frameworks on most datasets. The method is available under: https://github.com/s-marton/GRANDE

## 1 INTRODUCTION

Heterogeneous tabular data is the most frequently used form of data (Chui et al., 2018; Shwartz-Ziv & Armon, 2022) and is indispensable in a wide range of applications such as medical diagnosis (Ulmer et al., 2020; Somani et al., 2021), estimation of creditworthiness (Clements et al., 2020) and fraud detection (Cartella et al., 2021). Therefore, enhancing the predictive performance and robustness of models can bring significant advantages to users and companies (Borisov et al., 2022). However, tabular data comes with considerable challenges like noise, missing values, class imbalance, and a combination of different feature types, especially categorical and numerical data. Despite the success of deep learning (DL) in various domains, recent studies indicate that tabular data still poses a major challenge and tree-based models like XGBoost and CatBoost outperform them in most cases (Borisov et al., 2022; Grinsztajn et al., 2022; Shwartz-Ziv & Armon, 2022). At the same time, employing end-to-end gradient-based training provides several advantages over traditional machine learning methods (Borisov et al., 2022). They offer a high level of flexibility by allowing an easy integration of arbitrary, differentiable loss functions tailored towards specific problems and support iterative training (Sahoo et al., 2017). Moreover, gradient-based methods can be incorporated easily into multimodal learning, with tabular data being one of several input types (Lichtenwalter et al., 2021; Pölsterl et al., 2021). Therefore, creating tabular-specific, gradient-based methods is a very active field of research and the need for well-performing methods is intense (Grinsztajn et al., 2022).

Recently, Marton et al. (2023) introduced GradTree, a novel approach that uses gradient descent to learn hard, axis-aligned decision trees (DTs). This is achieved by reformulating DTs to a dense representation and jointly optimizing all tree parameters using backpropagation with a straight-through (ST) operator. Learning hard, axis-aligned DTs with gradient descent allows combining the advan-

tageous inductive bias of tree-based methods with the flexibility of a gradient-based optimization. In this paper, we propose GRANDE, **GRA**die**N**t-Based **D**ecision Tree **E**nsembles, a novel approach for learning decision tree ensembles using end-to-end gradient descent. Similar to Marton et al. (2023), we use a dense representation for split nodes and the ST operator to deal with the non-differentiable nature of DTs. We build upon their approach, transitioning from individual trees to a weighted tree ensemble, while maintaining an efficient computation. As a result, GRANDE holds a significant advantage over existing gradient-based methods. Typically, DL methods are biased towards smooth solutions (Rahaman et al., 2019). As the target function in tabular datasets is usually not smooth, DL methods struggle to find these irregular functions. In contrast, models that are based on hard, axis aligned DTs learn piece-wise constant functions and therefore do not show such a bias (Grinsztajn et al., 2022). This important advantage is one inherent aspect of GRANDE, as it utilizes hard, axis-aligned DTs. This is a major difference to existing DL methods for hierarchical representations like NODE, where soft and oblique splits are used (Popov et al., 2019). Furthermore, we introduce instance-wise weighting in GRANDE. This allows learning appropriate representations for simple and complex rules within a single model, which increases the performance of the ensemble. Furthermore, we show that our instance-wise weighting has a positive impact on the local interpretability relative to other state-of-the-art methods. More specifically, our contributions are as follows:

- We extend GradTree (Marton et al., 2023) from individual trees to an end-to-end gradient-based tree ensemble, maintaining efficient computation (Section 3.1).
- We introduce softsign as a differentiable split function and show the advantage over commonly used alternatives (Section 3.2).
- We propose a novel weighting technique that emphasizes instance-wise estimator importance (Section 3.3).

We conduct an extensive evaluation on 19 binary classification tasks (Section 4) based on the predefined tabular benchmark proposed by Bischl et al. (2021). GRANDE outperforms existing methods for both, default and optimized hyperparameters. The performance difference to other methods is substantial on several datasets, making GRANDE an important extension to the existing repertoire of tabular data methods.

## 2 BACKGROUND: GRADIENT-BASED DECISION TREES

GRANDE builds on gradient-based decision trees (GradTree) at the level of individual trees in the ensemble. Hence, we summarize the relevant aspects and notation of GradTree in this section and refer to Marton et al. (2023) for a complete overview.

Traditionally, DTs involve nested concatenation of rules. In GradTree, DTs are formulated as arithmetic functions based on addition and multiplication to facilitate gradient-based learning. Thereby both, GradTree and GRANDE focus on learning fully-grown (i.e., complete, full) DTs which can be pruned post-hoc. A DT of depth $d$ is formulated with respect to its parameters as:

$$t(\boldsymbol{x}|\boldsymbol{\lambda}, \boldsymbol{\tau}, \boldsymbol{\iota}) = \sum_{l=0}^{2^d-1} \lambda_l \, \mathbb{L}(\boldsymbol{x}|l, \boldsymbol{\tau}, \boldsymbol{\iota}) \tag{1}$$

where $\mathbb{L}$ is a function that indicates whether a sample $\boldsymbol{x} \in \mathbb{R}^n$ belongs to a leaf $l$, $\boldsymbol{\lambda} \in \mathcal{C}^{2^d}$ denotes class membership for each leaf node, $\boldsymbol{\tau} \in \mathbb{R}^{2^d-1}$ represents split thresholds and $\boldsymbol{\iota} \in \mathbb{N}^{2^d-1}$ the feature index for each internal node.

To support a gradient-based optimization and ensure an efficient computation via matrix operations, a novel dense DT representation is introduced in GradTree. Traditionally, the feature index vector $\boldsymbol{\iota}$ is one-dimensional, but GradTree expands it into a matrix form. Specifically, this representation one-hot encodes the feature index, converting $\boldsymbol{\iota} \in \mathbb{R}^{2^d-1}$ into a matrix $\boldsymbol{I} \in \mathbb{R}^{(2^d-1) \times n}$. Similarly, for split thresholds, instead of a single value for all features, individual values for each feature are stored, leading to a matrix representation $\boldsymbol{T} \in \mathbb{R}^{(2^d-1) \times n}$. By enumerating the internal nodes in breadth-first order, we can redefine the indicator function $\mathbb{L}$ for a leaf $l$, resulting in

$$g(\boldsymbol{x}|\boldsymbol{\lambda}, T, I) = \sum_{l=0}^{2^d-1} \lambda_l \, \mathbb{L}(\boldsymbol{x}|l, \boldsymbol{T}, \boldsymbol{I}) \tag{2}$$

$$\text{where} \quad \mathbb{L}(\boldsymbol{x}|l, \boldsymbol{T}, \boldsymbol{I}) = \prod_{j=1}^{d} \left(1 - \mathfrak{p}(l, j)\right) \mathbb{S}(\boldsymbol{x}|\boldsymbol{I}_{\mathfrak{i}(l,j)}, \boldsymbol{T}_{\mathfrak{i}(l,j)}) + \mathfrak{p}(l, j) \left(1 - \mathbb{S}(\boldsymbol{x}|\boldsymbol{I}_{\mathfrak{i}(l,j)}, \boldsymbol{T}_{\mathfrak{i}(l,j)})\right) \quad (3)$$

Here, $\mathfrak{i}$ is the index of the internal node preceding a leaf node $l$ at a certain depth $j$ and $\mathfrak{p}$ indicates whether the left ($\mathfrak{p} = 0$) or the right branch ($\mathfrak{p} = 1$) was taken.

Typically, DTs use the Heaviside step function for splitting, which is non-differentiable. GradTree reformulates the split function to account for reasonable gradients:

$$\mathbb{S}(\boldsymbol{x}|\boldsymbol{\iota}, \boldsymbol{\tau}) = \lfloor S \left(\boldsymbol{\iota} \cdot \boldsymbol{x} - \boldsymbol{\iota} \cdot \boldsymbol{\tau}\right) \rceil \quad (4)$$

Where $S(z) = \frac{1}{1+e^{-z}}$ represents the logistic function, $\lfloor z \rceil$ stands for rounding a real number $z$ to the nearest integer and $\boldsymbol{a} \cdot \boldsymbol{b}$ denotes the dot product between two vectors $\boldsymbol{a}$ and $\boldsymbol{b}$. We further need to ensure that $\boldsymbol{\iota}$ is a one-hot encoded vector to account for axis-aligned splits. This is achieved by applying a hardmax transformation before calculating $\mathbb{S}$. Both rounding and hardmax operations are non-differentiable. To overcome this, GradTree employs the straight-through (ST) operator during backpropagation. This allows the model to use non-differentiable operations in the forward pass while ensuring gradient propagation in the backward pass.

## 3 GRANDE: GRADIENT-BASED DECISION TREE ENSEMBLES

One core contribution of this paper is the extension of GradTree to tree ensembles (Section 3.1). In Section 3.2 we propose softsign as a differentiable split function to propagate more reasonable gradients. Furthermore, we introduce an instance-wise weighting in Section 3.3 and regularization techniques in Section 3.4. As a result, GRANDE can be learned end-to-end with gradient descent, leveraging the potential and flexibility of a gradient-based optimization.

### 3.1 FROM DECISION TREES TO WEIGHTED TREE ENSEMBLES

One advantage of GRANDE over existing gradient-based methods is the inductive bias of axis-aligned splits for tabular data. Combining this property with an end-to-end gradient-based optimization is at the core of GRANDE. This is also a major difference to existing DL methods for hierarchical representations like NODE, where soft, oblique splits are used (Popov et al., 2019). Therefore, we can define GRANDE as

$$G(\boldsymbol{x}|\boldsymbol{\omega}, \boldsymbol{L}, \mathsf{T}, \mathsf{l}) = \sum_{e=0}^{E} \omega_e \, g(\boldsymbol{x}|\boldsymbol{L}_e, \mathsf{T}_e, \mathsf{l}_e) \quad (5)$$

where $E$ is the number of estimators in the ensemble and $\boldsymbol{\omega}$ is a weight vector. By extending $\boldsymbol{L}$ to a matrix and $\mathsf{T}, \mathsf{l}$ to tensors for the complete ensemble instead of defining them individually for each tree, we can leverage parallel computation for an efficient training.

As GRANDE can be learned end-to-end with gradient descent, we keep an important advantage over existing, non-gradient-based tree methods like XGBoost and CatBoost. Both, the sequential induction of the individual trees and the sequential combination of individual trees via boosting are greedy. This results in constraints on the search space and can favor overfitting, as highlighted by Marton et al. (2023). In contrast, GRANDE learns all parameters of the ensemble jointly and overcomes these limitations.

### 3.2 DIFFERENTIABLE SPLIT FUNCTIONS

The Heaviside step function, which is commonly used as split function in DTs, is non-differentiable. To address this challenge, various studies have proposed the employment of differentiable split functions. A predominant approach is the adoption of the sigmoid function, which facilitates soft decisions (Jordan & Jacobs, 1994; Irsoy et al., 2012; Frosst & Hinton, 2017). A more recent development in this field originated with the introduction of the entmax transformation (Peters et al., 2019). Researchers utilized a two-class entmax (entmoid) function to turn the decisions more sparse (Popov et al., 2019). Further, Chang et al. (2021) proposed a temperature annealing procedure to gradually turn the decisions hard. Marton et al. (2023) introduced an alternative method for generating hard splits by using a straight-through (ST) operator after a sigmoid split function to generate hard splits.

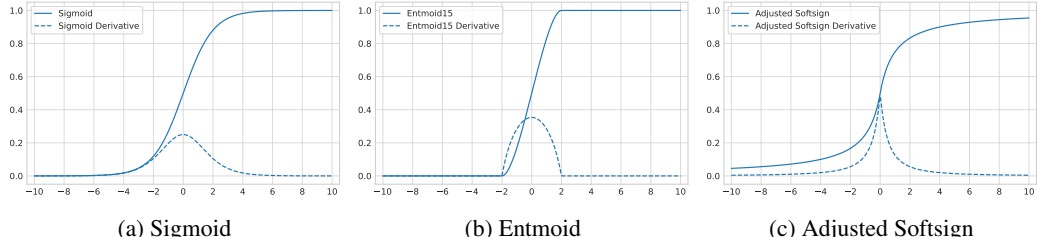

| (a) Sigmoid | (b) Entmoid | (c) Adjusted Softsign |

Figure 1: **Differentiable Split Functions.** The sigmoid gradient declines smoothly, while entmoid's gradient decays more rapidly but becomes zero for large values. The scaled softsign has high gradients for small values but maintains a responsive gradient for large values, offering greater sensitivity.

While this allows using hard splits for calculating the function values, it also introduces a mismatch between the forward and backward pass. However, we can utilize this to incorporate additional information: By using a sigmoid function, the distance between a feature value and the threshold is used as additional information during gradient computation. Accordingly, the gradient behavior plays a pivotal role in ensuring effective differentiation, especially in scenarios where input values are close to the decision threshold. The traditional sigmoid function can be suboptimal due to its smooth gradient decline. Entmoid, although addressing certain limitations of sigmoid, still displays an undesirable gradient behavior. Specifically, its gradient drops to zero when the difference in values is too pronounced. This can hinder the model's ability to accommodate samples that exhibit substantial variances from the threshold. Therefore, we propose using a softsign function, scaled to $(0, 1)$, as a differentiable split function:

$$S_{\text{ss}}(z) = \frac{1}{2} \left( \frac{z}{1 + |z|} + 1 \right) \tag{6}$$

The distinct gradient characteristics of the softsign, which are pronounced if samples are close to the threshold, reduce sharply but maintain responsive gradients if there is a large difference between the feature value and the threshold. These characteristics make it superior for differentiable splitting. This concept is visualized in Figure 1. Besides the intuitive advantage of using a softsign split function, we also show empirically that this is the superior choice (Table 4).

### 3.3 INSTANCE-WISE ESTIMATOR WEIGHTS

One challenge of ensemble methods is learning a good weighting scheme of the individual estimators. The flexibility of an end-to-end gradient-based optimization allows including learnable weight parameters to the optimization. A simple solution would be learning one weight for each estimator and using a softmax over all weights, resulting in a weighted average. However, this forces a very homogeneous ensemble, in which each tree aims to make equally good predictions for all samples. In contrast, it would be beneficial if individual trees can account for different areas of the target function, and are not required to make confident predictions for each sample.

To address this, we propose an advanced weighting scheme that allows calculating instance-wise weights that can be learned within the gradient-based optimization. Instead of using one weight per *estimator*, we use one weight for each *leaf* of the estimator as visualized in Figure 2 and thus define the weights as $\boldsymbol{W} \in \mathbb{R}^{E \times 2^d}$ instead of $\boldsymbol{\omega} \in \mathbb{R}^E$. We define $p(\boldsymbol{x}|\boldsymbol{L}, \mathsf{T}, \mathsf{l}) : \mathbb{R}^n \to \mathbb{R}^E$ as a function to calculate a vector comprising the individual prediction of each tree. Further, we define a function $w(\boldsymbol{x}|\boldsymbol{W}, \boldsymbol{L}, \mathsf{T}, \mathsf{l}) : \mathbb{R}^n \to \mathbb{R}^E$ to calculate a weight vector with one weight for each tree based on the leaf which the current sample is assigned to. Subsequently, a softmax is applied on these chosen weights for each sample. The process of multiplying the post-softmax weights by the predicted values from each tree equates to computing a weighted average. This results in

$$G(\boldsymbol{x}|\boldsymbol{W}, \boldsymbol{L}, \mathsf{T}, \mathsf{l}) = \sigma\left(w(\boldsymbol{x}|\boldsymbol{W}, \boldsymbol{L}, \mathsf{T}, \mathsf{l})\right) \cdot p(\boldsymbol{x}|\boldsymbol{L}, \mathsf{T}, \mathsf{l}) \tag{7}$$

$$\text{where} \quad w(\boldsymbol{x}|\boldsymbol{W}, \boldsymbol{L}, \mathsf{T}, \mathsf{l}) = \begin{bmatrix} \sum_l^{2^{d-1}} \boldsymbol{W}_{0,l}\, \mathbb{L}(\boldsymbol{x}|\boldsymbol{L}_{0,l}, \mathsf{T}_0, \mathsf{l}_0) \\ \sum_l^{2^{d-1}} \boldsymbol{W}_{1,l}\, \mathbb{L}(\boldsymbol{x}|\boldsymbol{L}_{1,l}, \mathsf{T}_1, \mathsf{l}_1) \\ \vdots \\ \sum_l^{2^{d-1}} \boldsymbol{W}_{E,l}\, \mathbb{L}(\boldsymbol{x}|\boldsymbol{L}_{E,l}, \mathsf{T}_E, \mathsf{l}_E) \end{bmatrix}, \quad p(\boldsymbol{x}|\boldsymbol{L}, \mathsf{T}, \mathsf{l}) = \begin{bmatrix} g(\boldsymbol{x}|\boldsymbol{L}_0, \mathsf{T}_0, \mathsf{l}_0) \\ g(\boldsymbol{x}|\boldsymbol{L}_1, \mathsf{T}_1, \mathsf{l}_1) \\ \vdots \\ g(\boldsymbol{x}|\boldsymbol{L}_E, \mathsf{T}_E, \mathsf{l}_E) \end{bmatrix}$$

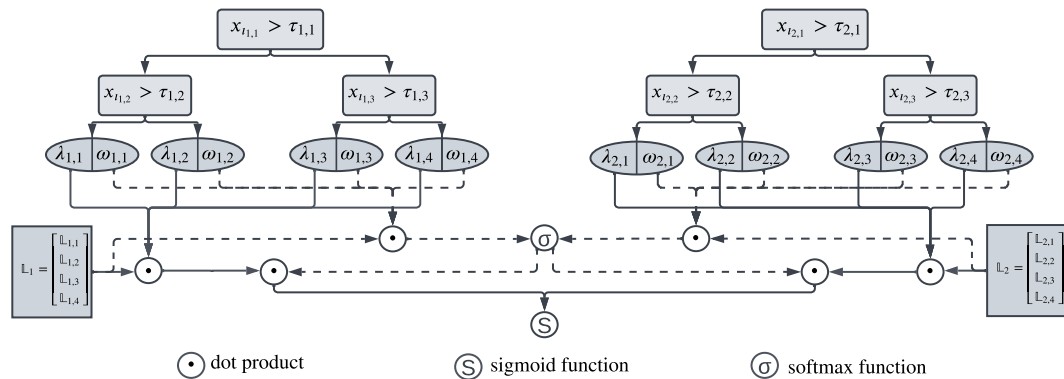

Figure 2: **GRANDE Architecture**. This figure visualizes the structure and weighting of GRANDE for an exemplary ensemble with two trees of depth two. For each tree in the ensemble, and for every sample, we determine the weight of the leaf which the sample is assigned to.

and $\sigma(\mathbf{z})$ is the softmax function. It is important to note that when calculating $\mathbb{L}$ (see Equation 3), only the value for the leaf to which the sample is assigned in a given tree is non-zero. We want to note that our weighting scheme permits calculating instance-wise weights even for unseen samples.

Our weighting scheme, in addition to its instance-wise nature, is substantially different to existing tree ensemble methods and post-hoc weighting schemes (He et al., 2014; Cui et al., 2023), as it is incorporated into the training procedure which is necessary to capture local interactions. Furthermore, the predictions of individual trees are not separate and changes in the instance-wise weights of one estimator directly impacts the weight of the remaining estimators. In our evaluation, we demonstrate that instance-wise weights significantly enhance the performance of GRANDE and emphasize local interpretability by learning representations for simple and complex rules within one model.

### 3.4 Regularization: Feature Subset, Data Subset and Dropout

The combination of tree-based methods with a gradient-based optimization opens the door for the application of numerous regularization techniques. For each tree in the ensemble, we select a feature subset. Therefore, we can regularize our model and simultaneously, we solve the poor scalability of GradTree with an increasing number of features. Similarly, we select a subset of the samples for each estimator. Furthermore, we implemented dropout by randomly deactivating a predefined fraction of the estimators in the ensemble and rescaling the weights accordingly.

### 4 Experimental Evaluation

As pointed out by Grinsztajn et al. (2022), most papers presenting a new method for tabular data have a highly varying evaluation methodology, with a small number of datasets that might be biased towards the authors' model. As a result, recent surveys showed that tree boosting methods like XGBoost and CatBoost are still state-of-the-art and outperform new architectures for tabular data on most datasets (Grinsztajn et al., 2022; Shwartz-Ziv & Armon, 2022; Borisov et al., 2022). This highlights the necessity for an extensive and unbiased evaluation, as we will carry out in the following, to accurately assess the performance of a new method and draw valid conclusions. We want to emphasize that recent surveys and evaluation on predefined benchmarks indicate that there is no "one-size-fits-all" solution for all tabular datasets. Consequently, we should view new methods as an extension to the existing repertoire and set our expectations in line with this perspective.

### 4.1 Experimental Setup

**Datasets and Preprocessing** For our evaluation, we used a predefined collection of datasets that was selected based on objective criteria from OpenML Benchmark Suites and comprises a total of 19 binary classification datasets (see Table 5 for details). The selection process was adopted from Bischl et al. (2021) and therefore is not biased towards our method. A more detailed discussion on

Table 2: **Performance Comparison.** We report the test macro F1-score (mean $\pm$ stdev for a 5-fold CV) with optimized parameters. The datasets are sorted based on the data size.

|  | GRANDE | XGB | CatBoost | NODE |
|---|---|---|---|---|
| dresses-sales | **0.612 $\pm$ 0.049 (1)** | 0.581 $\pm$ 0.059 (3) | 0.588 $\pm$ 0.036 (2) | 0.564 $\pm$ 0.051 (4) |
| climate-simulation-crashes | **0.853 $\pm$ 0.070 (1)** | 0.763 $\pm$ 0.064 (4) | 0.778 $\pm$ 0.050 (3) | 0.802 $\pm$ 0.035 (2) |
| cylinder-bands | **0.819 $\pm$ 0.032 (1)** | 0.773 $\pm$ 0.042 (3) | 0.801 $\pm$ 0.043 (2) | 0.754 $\pm$ 0.040 (4) |
| wdbc | **0.975 $\pm$ 0.010 (1)** | 0.953 $\pm$ 0.030 (4) | 0.963 $\pm$ 0.023 (3) | 0.966 $\pm$ 0.016 (2) |
| ilpd | **0.657 $\pm$ 0.042 (1)** | 0.632 $\pm$ 0.043 (3) | 0.643 $\pm$ 0.053 (2) | 0.526 $\pm$ 0.069 (4) |
| tokyo1 | 0.921 $\pm$ 0.004 (3) | 0.915 $\pm$ 0.011 (4) | **0.927 $\pm$ 0.013 (1)** | 0.921 $\pm$ 0.010 (2) |
| qsar-biodeg | **0.854 $\pm$ 0.022 (1)** | 0.853 $\pm$ 0.020 (2) | 0.844 $\pm$ 0.023 (3) | 0.836 $\pm$ 0.028 (4) |
| ozone-level-8hr | **0.726 $\pm$ 0.020 (1)** | 0.688 $\pm$ 0.021 (4) | 0.721 $\pm$ 0.027 (2) | 0.703 $\pm$ 0.029 (3) |
| madelon | 0.803 $\pm$ 0.010 (3) | 0.833 $\pm$ 0.018 (2) | **0.861 $\pm$ 0.012 (1)** | 0.571 $\pm$ 0.022 (4) |
| Bioresponse | 0.794 $\pm$ 0.008 (3) | 0.799 $\pm$ 0.011 (2) | **0.801 $\pm$ 0.014 (1)** | 0.780 $\pm$ 0.011 (4) |
| wilt | 0.936 $\pm$ 0.015 (2) | 0.911 $\pm$ 0.010 (4) | 0.919 $\pm$ 0.007 (3) | **0.937 $\pm$ 0.017 (1)** |
| churn | 0.914 $\pm$ 0.017 (2) | 0.900 $\pm$ 0.017 (3) | 0.869 $\pm$ 0.021 (4) | **0.930 $\pm$ 0.011 (1)** |
| phoneme | 0.846 $\pm$ 0.008 (4) | 0.872 $\pm$ 0.007 (2) | **0.876 $\pm$ 0.005 (1)** | 0.862 $\pm$ 0.013 (3) |
| SpeedDating | **0.723 $\pm$ 0.013 (1)** | 0.704 $\pm$ 0.015 (4) | 0.718 $\pm$ 0.014 (2) | 0.707 $\pm$ 0.015 (3) |
| PhishingWebsites | **0.969 $\pm$ 0.006 (1)** | 0.968 $\pm$ 0.006 (2) | 0.965 $\pm$ 0.003 (4) | 0.968 $\pm$ 0.006 (3) |
| Amazon_employee_access | 0.665 $\pm$ 0.009 (2) | 0.621 $\pm$ 0.008 (4) | **0.671 $\pm$ 0.011 (1)** | 0.649 $\pm$ 0.009 (3) |
| nomao | 0.958 $\pm$ 0.002 (3) | **0.965 $\pm$ 0.003 (1)** | 0.964 $\pm$ 0.002 (2) | 0.956 $\pm$ 0.001 (4) |
| adult | 0.790 $\pm$ 0.006 (4) | **0.798 $\pm$ 0.004 (1)** | 0.796 $\pm$ 0.004 (2) | 0.794 $\pm$ 0.004 (3) |
| numerai28.6 | **0.519 $\pm$ 0.003 (1)** | 0.518 $\pm$ 0.001 (3) | 0.519 $\pm$ 0.002 (2) | 0.503 $\pm$ 0.010 (4) |
| Normalized Mean $\uparrow$ | **0.776 (1)** | 0.483 (3) | 0.671 (2) | 0.327 (4) |
| Mean Reciprocal Rank (MRR) $\uparrow$ | **0.702 (1)** | 0.417 (3) | 0.570 (2) | 0.395 (4) |

the selection of the benchmark can be found in Appendix A. We one-hot encoded low-cardinality categorical features and used leave-one-out encoding for high-cardinality categorical features (more than 10 categories). To make them suitable for a gradient-based optimization, we gaussianized features using a quantile transformation, as it is common practice (Grinsztajn et al., 2022). In line with Borisov et al. (2022), we report the mean and standard deviation of the test performance over a 5-fold cross-validation to ensure reliable results.

**Methods** We compare our approach to XGBoost and CatBoost, which achieved superior results according to recent studies, and NODE, which is most related to our approach. With this setup, we have one state-of-the-art tree-based and

Table 1: **Categorization of Approaches**

|  | **Standard DTs** | **Oblivious DTs** |
|---|---|---|
| **Tree-based** | XGBoost | CatBoost |
| **Gradient-based** | GRANDE | NODE |

one gradient-based approach for each tree type (see Table 1). In addition, we provide an extended evaluation including SAINT, RandomForest and ExtraTree as additional benchmarks in Appendix B. These additional results are in line with the results presented in the following.

**Hyperparameters** We optimized the hyperparameters using Optuna (Akiba et al., 2019) with 250 trials and selected the search space as well as the default parameters for related work in accordance with Borisov et al. (2022). The best parameters were selected based on a 5x2 cross-validation as suggested by Raschka (2018) where the test data of each fold was held out of the HPO to get unbiased results. To deal with class imbalance, we further included class weights. Additional information along with the hyperparameters for each approach are in Appendix E.

## 4.2 RESULTS

**GRANDE outperforms existing methods on most datasets** We evaluated the performance with optimized hyperparameters based on the macro F1-Score in Table 2 to account for class imbalance. Additionally, we report the accuracy and ROC-AUC score in the Appendix B, which are consistent with the results presented in the following. GRANDE outperformed existing methods and achieved the highest mean reciprocal rank (MRR) of 0.702 and the highest normalized mean of 0.776. CatBoost yielded the second-best results (MRR of 0.570 and normalized mean of 0.671) followed by XGBoost (MRR of 0.417 and normalized mean of 0.483) and NODE (MRR of 0.395 and normalized mean of 0.327). Yet, our findings are in line with existing work, indicating that there is no universal method for tabular data. However, on several datasets such as *climate-simulation-crashes*

Table 4: **Ablation Study Summary.** Left: Comparison of different options for differentiable split functions (complete results in Table 10). Right: Comparison of our instance-wise weighting based on leaf weights with a single weight for each estimator (complete results in Table 11).

| | Differentiable Split Function | | | Weighting Technique | |
|---|---|---|---|---|---|
| | Softsign | Entmoid | Sigmoid | Leaf Weights | Estimator Weights |
| Normalized Mean ↑ | **0.7906 (1)** | 0.4188 (2) | 0.2207 (3) | **0.8235 (1)** | 0.1765 (2) |
| Mean Reciprocal Rank (MRR) ↑ | **0.8246 (1)** | 0.5526 (2) | 0.4561 (3) | **0.9211 (1)** | 0.5789 (2) |

and *cylinder-bands* the performance difference to other methods was substantial, which highlights the importance of GRANDE as an extension to the existing repertoire. Furthermore, as the datasets are sorted by their size, we can observe that the results of GRANDE are especially good for small datasets, which is an interesting research direction for future work.

**GRANDE efficient for large and high-dimensional datasets** GRANDE averaged 47 seconds across all datasets, with a maximum runtime of 107 seconds. Thereby, the runtime of GRANDE is robust to high-dimensional (37 seconds for *Bioresponse* with 1,776 features) and larger datasets (39 seconds for *numerai28.6* with 96,320 samples). GRANDE achieved a significantly lower runtime compared to our gradient-based benchmark NODE, which has an approximately three times higher average runtime of 130 seconds. However, it is important to note that GBDT frameworks, especially XGBoost, are highly efficient when executed on the GPU and achieve significantly lower runtimes compared to gradient-based methods. The complete runtimes are listed in the appendix (Table 9).

**GRANDE outperforms existing methods with default hyperparameters** Many methods, especially DL methods, are heavily reliant on a proper hyperparameter optimization. Yet, it is a desirable property that a method achieves good results even with their default setting. GRANDE achieves superior results with default hyperparameters, and significantly outperforms existing methods on most datasets. More specifically, GRANDE has the highest normalized mean performance (0.6371) and the highest MRR (0.6404) as summarized in Table 3.

Table 3: **Default Hyperparameter Performance Summary.** The results are based on the test macro F1-score with the default setting. Complete results are listed in Table 8.

| | Normalized Mean ↑ | Mean Reciprocal Rank (MRR) ↑ |
|---|---|---|
| GRANDE | **0.6371 (1)** | **0.6404 (1)** |
| XGB | 0.5865 (2) | 0.5175 (3) |
| CatBoost | 0.5793 (3) | 0.5219 (2) |
| NODE | 0.2698 (4) | 0.4035 (4) |

**Softsign improves performance** As discussed in Section 3.2, we argue that employing softsign as split index activation propagates informative gradients beneficial for the optimization. In Table 4 we support these claims by showing a superior performance of GRANDE with a softsign activation (before discretizing with the ST operator) compared to sigmoid as the default choice as well as an entmoid function which is commonly used in related work (Popov et al., 2019; Chang et al., 2021).

**Instance-wise weighting increases model performance** GRANDE uses instance-wise weighting to assign varying weights to estimators for each sample based on selected leaves. This promotes ensemble diversity and encourages estimators to capture unique local interactions. We argue that the ability to learn and represent simple, local rules with individual estimators in our ensemble can have a positive impact on the overall performance as it simplifies the task that has to be solved by the remaining estimators. As a result, GRANDE can efficiently learn compact representations for simple rules, where complex models usually tend to learn overly complex representations.

### 4.3 CASE STUDY: INSTANCE-WISE WEIGHTING FOR THE PHISHINGWEBSITES DATASET

In the following case study, we demonstrate the ability of GRANDE to learn compact representations for simple rules within a complex ensemble: The *PhishingWebsites* dataset is concerned with identifying malicious websites based on metadata and additional observable characteristics. Although the task is challenging (i.e., it is not possible to solve it sufficiently well with a simple model, as shown in Table 12), there exist several clear indicators for phishing websites. Thus, some instances can be categorized using simple rules, while assigning other instances is more difficult. Ideally, if an instance can be easily categorized, the model should follow simple rules to make a prediction.

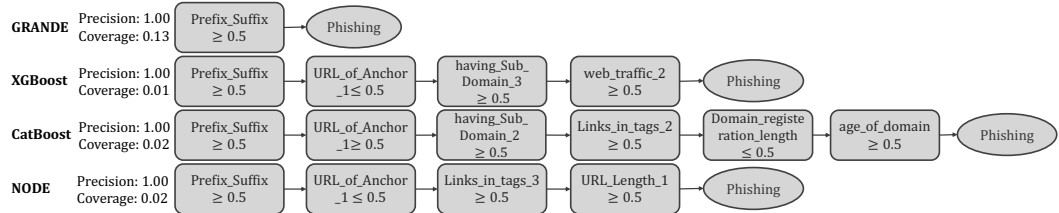

Figure 4: **Anchors Explanations.** This figure shows the local explanations generated by Anchors for the given instance. The explanation for GRANDE only comprises a single rule. In contrast, the corresponding explanations for the other methods have significantly higher complexity, which indicates that these methods are not able to learn simple representations within a complex model.

One example for a rule, which holds universally in the given dataset, is that an instance can be classified as *phishing* if a prefix or suffix was added to the domain name. By assessing the weights for an exemplary instance fulfilling this rule, we can observe that the DT visualized in Figure 3 accounts for 94% of the prediction. Accordingly, GRANDE has learned a very simple representation and the classification is derived by applying an easily comprehensible rule. Notably, for the other methods, it is not possible to assess the importance of individual estimators out-of-the-box similarly, as the prediction is either derived by either sequentially summing up the predictions (e.g. XGBoost and CatBoost) or equally weighting all estimators. Furthermore, this has a significant positive impact on the average performance of GRANDE compared to using one single weight for each estimator (see Table 4).

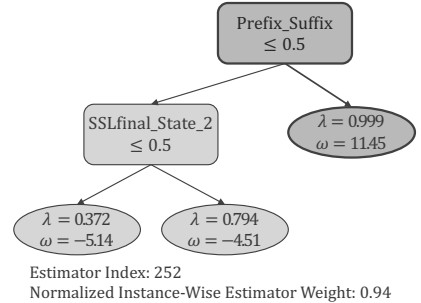

Estimator Index: 252
Normalized Instance-Wise Estimator Weight: 0.94

Figure 3: **Highest-Weighted Estimator**. This figure visualizes the DT from GRANDE (1024 total estimators) which has the highest weight for an exemplary instance.

**Instance-wise weighting can be beneficial for local interpretability** In addition to the performance increase, our instance-wise weighting has a notable impact on the local interpretability of GRANDE. For each instance, we can assess the weights of individual estimators and inspect the estimators with the highest importance to understand which rules have the greatest impact on the prediction. For the given example, we only need to observe a single tree of depth two (Figure 3) to understand why the given instance was classified as *phishing*, even though the complete model is very complex. In contrast, existing ensemble methods require a global interpretation of the model and do not provide simple, local explanations out-of-the-box.

However, similar explanations can be extracted using Anchors (Ribeiro et al., 2018). Anchors, as an extension to LIME (Ribeiro et al., 2016), provides model-agnostic explanations by identifying conditions (called "anchors") which, when satisfied, guarantee a certain prediction with a high probability (noted as precision). These anchors are interpretable, rules-based conditions derived from input features that consistently lead to the same model prediction. Figure 4 shows the extracted rules for each approach. We can clearly see that the anchor extracted for GRANDE matches the rule we have identified based on the instance-wise weights in Figure 3. Furthermore, it is evident that the prediction derived by GRANDE is much simpler compared to any other approach, as it only comprises a single rule. Notably, this comes without suffering a loss in the precision, which is 1.00 for all methods. Furthermore, the rule learned by GRANDE has a significantly higher coverage, which means that the rule applied by GRANDE is more broadly representative. The corresponding experiment with additional details can be found in the supplementary material, and a more detailed discussion of the weighting statistics is included in Appendix D.

## 5 RELATED WORK

Tabular data is the most frequently used type of data, and learning methods for tabular data are a field of very active research. Existing work can be divided into tree-based, DL and hybrid methods.

In the following, we categorize the most prominent methods based on these three categories and differentiate our approach from existing work. For a more comprehensive review, we refer to Borisov et al. (2022), Shwartz-Ziv & Armon (2022) and Grinsztajn et al. (2022).

**Tree-Based Methods**    Tree-based methods have been widely used for tabular data due to their interpretability and ability to capture non-linear relationships. While individual trees usually offer a higher interpretability, tree ensemble methods Breiman (2001); Geurts et al. (2006), most notably gradient-boosted DTs (GBDT) are commonly used to achieve superior performance (Friedman, 2001). The most prominent GBDT methods for tabular data improve the gradient boosting algorithm by for instance introducing advanced regularization (XGBoost (Chen & Guestrin, 2016)), a special handling for categorical variables (CatBoost (Prokhorenkova et al., 2018)) or a leaf-wise growth strategy (LightGBM (Ke et al., 2017)). Regarding the structure, GRANDE is similar to existing tree-based models. The main difference is the end-to-end gradient-based training procedure, which offers additional flexibility, and the instance-wise weighting.

**Deep Learning Methods**    With the success of DL in various domains, researchers have started to adjust DL architectures, mostly transformers, to tabular data (Gorishniy et al., 2021; Arik & Pfister, 2021; Huang et al., 2020; Cai et al., 2021; Kossen et al., 2021). According to recent studies, Self-Attention and Intersample Attention Transformer (SAINT) is the superior DL method for tabular data using attention over both, rows and columns (Somepalli et al., 2021). Although GRANDE, similar to DL methods, uses gradient descent for training, it has a shallow, hierarchical structure comprising hard, axis-aligned splits.

**Hybrid Methods**    Hybrid methods aim to combine the strengths of a gradient-based optimization with other algorithms, most commonly tree-based methods (Yang et al., 2018; Abutbul et al., 2020; Hehn et al., 2020; Chen, 2020; Ke et al., 2019; 2018; Katzir et al., 2020). One prominent way to achieve this is using soft DTs to apply gradient descent by replacing hard decisions with soft ones, and axis-aligned with oblique splits (Frosst & Hinton, 2017; Kontschieder et al., 2015; Luo et al., 2021; Hazimeh et al., 2020; Yu et al., 2021). Neural Oblivious Decision Ensembles (NODE) is one prominent hybrid method which learns ensembles of oblivious DTs with gradient descent and is therefore closely related to our work (Popov et al., 2019). Oblivious DTs use the same splitting feature and threshold for each internal node at the same depth, which allows an efficient, parallel computation and makes them suitable as weak learners. In contrast, GRANDE uses standard DTs as weak learners. GRANDE can also be categorized as a hybrid method. The main difference to existing methods is the use of hard, axis-aligned splits, which prevents overly smooth solution typically inherent in soft, oblique trees.

While some works demonstrate strong results of DL methods (Kadra et al., 2021), recent studies indicate that, despite huge effort in finding high-performant DL methods, tree-based models still outperform DL for tabular data (Grinsztajn et al., 2022; Borisov et al., 2022; Shwartz-Ziv & Armon, 2022), even though the gap is diminishing (McElfresh et al., 2023). One main reason for the superior performance of tree-based methods lies in the use of axis-aligned splits that are not biased towards overly smooth solutions (Grinsztajn et al., 2022). Therefore, GRANDE aligns with this argument and uses hard, axis-aligned splits combined with the flexibility of a gradient-based optimization.

## 6    CONCLUSION AND FUTURE WORK

In this paper, we introduced GRANDE, a new method for learning hard, axis-aligned tree ensembles with gradient-descent. GRANDE combines the advantageous inductive bias of axis-aligned splits with the flexibility offered by gradient descent optimization. In an extensive evaluation on a predefined benchmark, we demonstrated that GRANDE achieved superior results. Both with optimized and default parameters, it outperformed existing state-of-the-art methods on most datasets. Furthermore, we showed that the instance-wise weighting of GRANDE emphasizes learning representations for simple and complex relations within a single model, which increases the local interpretability compared to existing methods.

Currently, the proposed architecture is a shallow ensemble and already achieves state-of-the-art performance. However, the flexibility of a gradient-based optimization holds potential e.g., by including categorical embeddings, stacking of tree layers and an incorporation of tree layers to DL frameworks, which is subject to future work.

ACKNOWLEDGMENTS

This research was supported in part by the German Federal Ministry for Economic Affairs and Climate Action of Germany (BMWK), and in part by the German Federal Ministry for Environment, Nature Conservation and Nuclear Safety (BMUV).

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

## A  BENCHMARK DATASET SELCTION

We decided against using the original CC18 benchmark, as the number of datasets (72) is extremely high and as reported by Grinsztajn et al. (2022), the selection process was not strict enough, as there are many simple datasets contained. Similarly, we decided not to use the tabular benchmark presented by Grinsztajn et al. (2022), as the datasets were adjusted to be extremely homogenous by removing all side-aspects (e.g. class imbalance, high-dimensionality, dataset size). We argue that this also removes some main challenges when dealing with tabular data. As a result, we decided to use the benchmark proposed by Bischl et al. (2021)[1], which has a more strict selection process than CC18. The benchmark originally includes both, binary and multi-class tasks. For this paper, due to the limited scope, we focused only on binary classification tasks. Yet, our benchmark has a large overlap with CC18 as 16/19 datasets are also contained in CC18. The overlap with Grinsztajn et al. (2022) in contrast is rather small. This is mainly caused by the fact that most datasets in their tabular benchmark are binarized versions of multi-class or regression datasets, which was not allowed during the selection of our benchmark. Table 5 lists the used datasets, along with relevant statistics and the source based on the OpenML-ID.

Table 5: **Datasets**

|  | Samples | Features | Categorical Features | Features (preprocessed) | Minority Class | OpenML ID |
|---|---|---|---|---|---|---|
| dresses-sales | 500 | 12 | 11 | 37 | 42.00% | 23381 |
| climate-simulation-crashes | 540 | 18 | 0 | 18 | 8.52% | 40994 |
| cylinder-bands | 540 | 37 | 19 | 82 | 42.22% | 6332 |
| wdbc | 569 | 30 | 0 | 30 | 37.26% | 1510 |
| ilpd | 583 | 10 | 1 | 10 | 28.64% | 1480 |
| tokyo1 | 959 | 44 | 2 | 44 | 36.08% | 40705 |
| qsar-biodeg | 1,055 | 41 | 0 | 41 | 33.74% | 1494 |
| ozone-level-8hr | 2,534 | 72 | 0 | 72 | 6.31% | 1487 |
| madelon | 2,600 | 500 | 0 | 500 | 50.00% | 1485 |
| Bioresponse | 3,751 | 1,776 | 0 | 1,776 | 45.77% | 4134 |
| wilt | 4,839 | 5 | 0 | 5 | 5.39% | 40983 |
| churn | 5,000 | 20 | 4 | 22 | 14.14% | 40701 |
| phoneme | 5,404 | 5 | 0 | 5 | 29.35% | 1489 |
| SpeedDating | 8,378 | 120 | 61 | 241 | 16.47% | 40536 |
| PhishingWebsites | 11,055 | 30 | 30 | 46 | 44.31% | 4534 |
| Amazon_employee_access | 32,769 | 9 | 9 | 9 | 5.79% | 4135 |
| nomao | 34,465 | 118 | 29 | 172 | 28.56% | 1486 |
| adult | 48,842 | 14 | 8 | 37 | 23.93% | 1590 |
| numerai28.6 | 96,320 | 21 | 0 | 21 | 49.43% | 23517 |

## B  ADDITIONAL RESULTS

---

[1]The notebook for dataset selection can be accessed under `https://github.com/openml/benchmark-suites/blob/master/OpenML%20Benchmark%20generator.ipynb`.

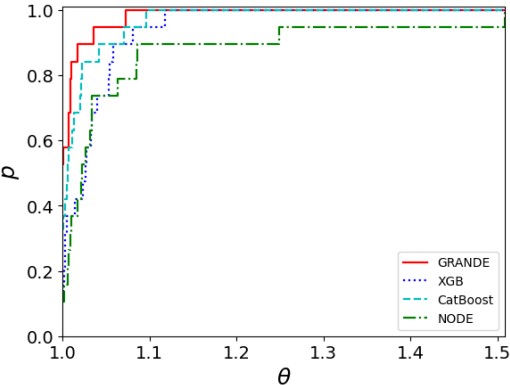

Figure 5: **Performance Profile (HPO 250 Trials)**. The performance profile is based on the macro F1-Score with optimized hyperparameters (complete grids, 250 trials). The x-axis represents a tolerance factor, and the y-axis is a proportion of the evaluated datasets.

Table 6: **ROC-AUC Performance Comparison (HPO 250 Trials).** We report the test ROC-AUC (mean $\pm$ stdev for a 5-fold CV) with optimized parameters (complete grids, 250 trials) and the ranking of each approach in parentheses. The datasets are sorted based on the data size.

| | GRANDE | XGB | CatBoost | NODE |
|---|---|---|---|---|
| dresses-sales | $0.623 \pm 0.049$ (2) | $0.589 \pm 0.060$ (4) | $0.607 \pm 0.040$ (3) | $\mathbf{0.632 \pm 0.041}$ **(1)** |
| climate-simulation-crashes | $\mathbf{0.958 \pm 0.031}$ **(1)** | $0.923 \pm 0.031$ (4) | $0.938 \pm 0.036$ (2) | $0.933 \pm 0.030$ (3) |
| cylinder-bands | $\mathbf{0.896 \pm 0.032}$ **(1)** | $0.872 \pm 0.019$ (3) | $0.879 \pm 0.046$ (2) | $0.837 \pm 0.037$ (4) |
| wdbc | $\mathbf{0.993 \pm 0.006}$ **(1)** | $0.990 \pm 0.010$ (4) | $0.990 \pm 0.010$ (3) | $0.993 \pm 0.007$ (2) |
| ilpd | $\mathbf{0.748 \pm 0.046}$ **(1)** | $0.721 \pm 0.047$ (4) | $0.728 \pm 0.055$ (3) | $0.745 \pm 0.048$ (2) |
| tokyo1 | $0.983 \pm 0.005$ (2) | $0.982 \pm 0.006$ (3) | $\mathbf{0.984 \pm 0.005}$ **(1)** | $0.980 \pm 0.005$ (4) |
| qsar-biodeg | $\mathbf{0.934 \pm 0.008}$ **(1)** | $0.925 \pm 0.008$ (3) | $0.933 \pm 0.011$ (2) | $0.920 \pm 0.009$ (4) |
| ozone-level-8hr | $\mathbf{0.925 \pm 0.013}$ **(1)** | $0.879 \pm 0.012$ (4) | $0.910 \pm 0.011$ (2) | $0.906 \pm 0.021$ (3) |
| madelon | $0.875 \pm 0.008$ (3) | $0.904 \pm 0.014$ (2) | $\mathbf{0.928 \pm 0.012}$ **(1)** | $0.612 \pm 0.016$ (4) |
| Bioresponse | $0.872 \pm 0.003$ (3) | $\mathbf{0.873 \pm 0.007}$ **(1)** | $0.873 \pm 0.002$ (2) | $0.859 \pm 0.008$ (4) |
| wilt | $0.994 \pm 0.007$ (2) | $0.981 \pm 0.015$ (4) | $0.991 \pm 0.009$ (3) | $\mathbf{0.996 \pm 0.003}$ **(1)** |
| churn | $\mathbf{0.928 \pm 0.014}$ **(1)** | $0.919 \pm 0.018$ (4) | $0.920 \pm 0.013$ (3) | $0.927 \pm 0.014$ (2) |
| phoneme | $0.939 \pm 0.006$ (3) | $0.955 \pm 0.007$ (2) | $\mathbf{0.959 \pm 0.005}$ **(1)** | $0.934 \pm 0.010$ (4) |
| SpeedDating | $\mathbf{0.859 \pm 0.012}$ **(1)** | $0.827 \pm 0.017$ (4) | $0.856 \pm 0.014$ (2) | $0.853 \pm 0.014$ (3) |
| PhishingWebsites | $0.996 \pm 0.001$ (2) | $\mathbf{0.996 \pm 0.001}$ **(1)** | $0.996 \pm 0.001$ (4) | $0.996 \pm 0.001$ (3) |
| Amazon_employee_access | $0.830 \pm 0.010$ (3) | $0.778 \pm 0.015$ (4) | $\mathbf{0.842 \pm 0.014}$ **(1)** | $0.841 \pm 0.009$ (2) |
| nomao | $0.994 \pm 0.001$ (3) | $\mathbf{0.996 \pm 0.001}$ **(1)** | $0.995 \pm 0.001$ (2) | $0.993 \pm 0.001$ (4) |
| adult | $0.910 \pm 0.005$ (4) | $\mathbf{0.927 \pm 0.002}$ **(1)** | $0.925 \pm 0.003$ (2) | $0.915 \pm 0.003$ (3) |
| numerai28.6 | $0.529 \pm 0.003$ (3) | $0.529 \pm 0.002$ (2) | $\mathbf{0.529 \pm 0.002}$ **(1)** | $0.529 \pm 0.003$ (4) |
| Normalized Mean $\uparrow$ | **0.769 (1)** | 0.419 (3) | 0.684 (2) | 0.395 (4) |
| Mean Reciprocal Rank (MRR) $\uparrow$ | **0.645 (1)** | 0.461 (3) | 0.575 (2) | 0.404 (4) |

Table 7: **Accuracy Performance Comparison (HPO 250 Trials).** We report the test balanced accuracy (mean ± stdev for a 5-fold CV) with optimized parameters (complete grids, 250 trials) and the ranking of each approach in parentheses. The datasets are sorted based on the data size.

| | GRANDE | XGB | CatBoost | NODE |
|---|---|---|---|---|
| dresses-sales | **0.613 ± 0.048 (1)** | 0.585 ± 0.060 (3) | 0.589 ± 0.037 (2) | 0.581 ± 0.037 (4) |
| climate-simulation-crashes | **0.875 ± 0.079 (1)** | 0.762 ± 0.060 (4) | 0.832 ± 0.035 (2) | 0.773 ± 0.052 (3) |
| cylinder-bands | **0.817 ± 0.032 (1)** | 0.777 ± 0.045 (3) | 0.798 ± 0.042 (2) | 0.751 ± 0.040 (4) |
| wdbc | **0.973 ± 0.010 (1)** | 0.952 ± 0.029 (4) | 0.963 ± 0.023 (3) | 0.963 ± 0.017 (2) |
| ilpd | **0.709 ± 0.046 (1)** | 0.673 ± 0.042 (3) | 0.689 ± 0.062 (2) | 0.548 ± 0.054 (4) |
| tokyo1 | 0.925 ± 0.002 (2) | 0.918 ± 0.012 (4) | **0.932 ± 0.010 (1)** | 0.920 ± 0.007 (3) |
| qsar-biodeg | 0.853 ± 0.024 (2) | **0.856 ± 0.022 (1)** | 0.847 ± 0.028 (3) | 0.831 ± 0.032 (4) |
| ozone-level-8hr | **0.774 ± 0.016 (1)** | 0.733 ± 0.021 (3) | 0.735 ± 0.034 (2) | 0.669 ± 0.033 (4) |
| madelon | 0.803 ± 0.010 (3) | 0.833 ± 0.018 (2) | **0.861 ± 0.012 (1)** | 0.571 ± 0.022 (4) |
| Bioresponse | 0.795 ± 0.009 (3) | 0.799 ± 0.011 (2) | **0.801 ± 0.014 (1)** | 0.780 ± 0.011 (4) |
| wilt | **0.962 ± 0.026 (1)** | 0.941 ± 0.012 (4) | 0.955 ± 0.021 (2) | 0.948 ± 0.024 (3) |
| churn | **0.909 ± 0.014 (1)** | 0.895 ± 0.022 (3) | 0.894 ± 0.014 (4) | 0.904 ± 0.021 (2) |
| phoneme | 0.859 ± 0.011 (4) | 0.882 ± 0.005 (2) | **0.886 ± 0.006 (1)** | 0.859 ± 0.013 (3) |
| SpeedDating | 0.752 ± 0.020 (2) | 0.740 ± 0.017 (3) | **0.758 ± 0.012 (1)** | 0.694 ± 0.015 (4) |
| PhishingWebsites | **0.969 ± 0.006 (1)** | 0.968 ± 0.006 (2) | 0.965 ± 0.003 (4) | 0.968 ± 0.006 (3) |
| Amazon_employee_access | 0.707 ± 0.010 (2) | 0.701 ± 0.015 (3) | **0.775 ± 0.009 (1)** | 0.617 ± 0.007 (4) |
| nomao | 0.961 ± 0.001 (3) | 0.969 ± 0.002 (2) | **0.969 ± 0.001 (1)** | 0.956 ± 0.001 (4) |
| adult | 0.817 ± 0.008 (3) | 0.841 ± 0.004 (2) | **0.841 ± 0.004 (1)** | 0.778 ± 0.008 (4) |
| numerai28.6 | 0.520 ± 0.004 (2) | 0.520 ± 0.000 (3) | **0.521 ± 0.002 (1)** | 0.519 ± 0.003 (4) |
| Normalized Mean ↑ | **0.793 (1)** | 0.523 (3) | 0.730 (2) | 0.126 (4) |
| Mean Reciprocal Rank (MRR) ↑ | 0.689 (2) | 0.404 (3) | **0.693 (1)** | 0.298 (4) |

Table 8: **Default Parameter Performance Comparison.** We report the test macro F1-score (mean ± stdev over 10 trials) with default parameters and the ranking of each approach in parentheses. The datasets are sorted based on the data size.

| | GRANDE | XGB | CatBoost | NODE |
|---|---|---|---|---|
| dresses-sales | **0.596 ± 0.014 (1)** | 0.570 ± 0.056 (3) | 0.573 ± 0.031 (2) | 0.559 ± 0.045 (4) |
| climate-simulation-crashes | 0.758 ± 0.065 (4) | **0.781 ± 0.060 (1)** | 0.781 ± 0.050 (2) | 0.766 ± 0.088 (3) |
| cylinder-bands | **0.813 ± 0.023 (1)** | 0.770 ± 0.010 (3) | 0.795 ± 0.051 (2) | 0.696 ± 0.028 (4) |
| wdbc | 0.962 ± 0.008 (3) | **0.966 ± 0.023 (1)** | 0.955 ± 0.029 (4) | 0.964 ± 0.017 (2) |
| ilpd | **0.646 ± 0.021 (1)** | 0.629 ± 0.052 (3) | 0.643 ± 0.042 (2) | 0.501 ± 0.085 (4) |
| tokyo1 | **0.922 ± 0.014 (1)** | 0.917 ± 0.016 (4) | 0.917 ± 0.013 (3) | 0.921 ± 0.011 (2) |
| qsar-biodeg | **0.851 ± 0.032 (1)** | 0.844 ± 0.021 (2) | 0.843 ± 0.017 (3) | 0.838 ± 0.027 (4) |
| ozone-level-8hr | **0.735 ± 0.011 (1)** | 0.686 ± 0.034 (3) | 0.702 ± 0.029 (2) | 0.662 ± 0.019 (4) |
| madelon | 0.768 ± 0.022 (3) | 0.811 ± 0.016 (2) | **0.851 ± 0.015 (1)** | 0.650 ± 0.017 (4) |
| Bioresponse | 0.789 ± 0.014 (2) | 0.789 ± 0.013 (3) | **0.792 ± 0.004 (1)** | 0.786 ± 0.010 (4) |
| wilt | **0.933 ± 0.021 (1)** | 0.903 ± 0.011 (3) | 0.898 ± 0.011 (4) | 0.904 ± 0.026 (2) |
| churn | 0.896 ± 0.007 (3) | 0.897 ± 0.022 (2) | 0.862 ± 0.015 (4) | **0.925 ± 0.025 (1)** |
| phoneme | 0.860 ± 0.008 (3) | **0.864 ± 0.003 (1)** | 0.861 ± 0.008 (2) | 0.842 ± 0.005 (4) |
| SpeedDating | **0.725 ± 0.007 (1)** | 0.686 ± 0.010 (4) | 0.693 ± 0.013 (3) | 0.703 ± 0.013 (2) |
| PhishingWebsites | **0.969 ± 0.006 (1)** | 0.969 ± 0.007 (2) | 0.963 ± 0.005 (3) | 0.961 ± 0.004 (4) |
| Amazon_employee_access | 0.602 ± 0.006 (4) | 0.608 ± 0.016 (3) | **0.652 ± 0.006 (1)** | 0.621 ± 0.010 (2) |
| nomao | 0.955 ± 0.004 (3) | **0.965 ± 0.003 (1)** | 0.962 ± 0.003 (2) | 0.955 ± 0.002 (4) |
| adult | 0.785 ± 0.008 (4) | 0.796 ± 0.003 (2) | 0.796 ± 0.005 (3) | **0.799 ± 0.003 (1)** |
| numerai28.6 | 0.503 ± 0.003 (4) | 0.516 ± 0.002 (2) | **0.519 ± 0.001 (1)** | 0.506 ± 0.009 (3) |
| Normalized Mean ↑ | **0.637 (1)** | 0.587 (3) | 0.579 (1) | 0.270 (4) |
| Mean Reciprocal Rank (MRR) ↑ | **0.640 (1)** | 0.518 (3) | 0.522 (2) | 0.404 (4) |

Table 9: **Runtime Performance Comparison.** We report the runtime (mean $\pm$ stdev for a 5-fold CV) with optimized parameters (complete grids, 250 trials) and the ranking of each approach in parentheses. The datasets are sorted based on the data size. For all methods, we used a single NVIDIA RTX A6000.

| | GRANDE | XGB | CatBoost | NODE |
|---|---|---|---|---|
| dresses-sales | 11.121 $\pm$ 1.0 (3) | **0.052 $\pm$ 0.0 (1)** | 4.340 $\pm$ 2.0 (2) | 31.371 $\pm$ 17.0 (4) |
| climate-simulation-crashes | 16.838 $\pm$ 3.0 (3) | **0.077 $\pm$ 0.0 (1)** | 8.740 $\pm$ 9.0 (2) | 97.693 $\pm$ 10.0 (4) |
| cylinder-bands | 20.554 $\pm$ 2.0 (3) | **0.093 $\pm$ 0.0 (1)** | 4.887 $\pm$ 2.0 (2) | 38.983 $\pm$ 12.0 (4) |
| wdbc | 29.704 $\pm$ 4.0 (3) | **0.151 $\pm$ 0.0 (1)** | 1.046 $\pm$ 0.0 (2) | 83.548 $\pm$ 16.0 (4) |
| ilpd | 11.424 $\pm$ 1.0 (3) | **0.049 $\pm$ 0.0 (1)** | 3.486 $\pm$ 3.0 (2) | 59.085 $\pm$ 24.0 (4) |
| tokyo1 | 21.483 $\pm$ 3.0 (3) | **0.078 $\pm$ 0.0 (1)** | 1.485 $\pm$ 1.0 (2) | 84.895 $\pm$ 5.0 (4) |
| qsar-biodeg | 21.565 $\pm$ 2.0 (3) | **0.087 $\pm$ 0.0 (1)** | 1.195 $\pm$ 0.0 (2) | 96.204 $\pm$ 20.0 (4) |
| ozone-level-8hr | 56.889 $\pm$ 6.0 (3) | **0.092 $\pm$ 0.0 (1)** | 0.851 $\pm$ 0.0 (2) | 137.910 $\pm$ 27.0 (4) |
| madelon | 44.783 $\pm$ 24.0 (3) | **0.360 $\pm$ 0.0 (1)** | 1.247 $\pm$ 0.0 (2) | 90.529 $\pm$ 13.0 (4) |
| Bioresponse | 37.224 $\pm$ 2.0 (3) | **0.865 $\pm$ 0.0 (1)** | 2.136 $\pm$ 0.0 (2) | 309.178 $\pm$ 54.0 (4) |
| wilt | 44.476 $\pm$ 7.0 (3) | **0.127 $\pm$ 0.0 (1)** | 1.090 $\pm$ 0.0 (2) | 199.653 $\pm$ 20.0 (4) |
| churn | 49.096 $\pm$ 5.0 (3) | **0.099 $\pm$ 0.0 (1)** | 4.117 $\pm$ 3.0 (2) | 150.088 $\pm$ 33.0 (4) |
| phoneme | 59.286 $\pm$ 7.0 (3) | **0.201 $\pm$ 0.0 (1)** | 1.793 $\pm$ 1.0 (2) | 240.607 $\pm$ 33.0 (4) |
| SpeedDating | 83.458 $\pm$ 24.0 (4) | **0.207 $\pm$ 0.0 (1)** | 7.033 $\pm$ 1.0 (2) | 66.560 $\pm$ 22.0 (3) |
| PhishingWebsites | 107.101 $\pm$ 37.0 (3) | **0.271 $\pm$ 0.0 (1)** | 8.527 $\pm$ 2.0 (2) | 340.660 $\pm$ 102.0 (4) |
| Amazon_employee_access | 37.190 $\pm$ 1.0 (4) | **0.047 $\pm$ 0.0 (1)** | 2.021 $\pm$ 0.0 (2) | 30.309 $\pm$ 4.0 (3) |
| nomao | 95.775 $\pm$ 11.0 (3) | **0.268 $\pm$ 0.0 (1)** | 10.911 $\pm$ 2.0 (2) | 208.682 $\pm$ 34.0 (4) |
| adult | 96.737 $\pm$ 6.0 (3) | **0.125 $\pm$ 0.0 (1)** | 3.373 $\pm$ 1.0 (2) | 171.783 $\pm$ 34.0 (4) |
| numerai28.6 | 39.031 $\pm$ 3.0 (3) | **0.083 $\pm$ 0.0 (1)** | 1.323 $\pm$ 1.0 (2) | 47.520 $\pm$ 38.0 (4) |
| Mean $\downarrow$ | 46.512 (3) | **0.175 (1)** | 3.66 (2) | 130.80 (4) |

Table 10: **Ablation Study Split Activation.** We report the test macro F1-Score (mean $\pm$ stdev for a 5-fold CV) with optimized parameters (complete grids, 250 trials) and the ranking of each approach in parentheses. The datasets are sorted based on the data size.

| | Softsign | Entmoid | Sigmoid |
|---|---|---|---|
| dresses-sales | **0.612 $\pm$ 0.049 (1)** | 0.580 $\pm$ 0.047 (2) | 0.568 $\pm$ 0.052 (3) |
| climate-simulation-crashes | **0.853 $\pm$ 0.070 (1)** | 0.840 $\pm$ 0.038 (2) | 0.838 $\pm$ 0.041 (3) |
| cylinder-bands | **0.819 $\pm$ 0.032 (1)** | 0.802 $\pm$ 0.028 (3) | 0.807 $\pm$ 0.020 (2) |
| wdbc | **0.975 $\pm$ 0.010 (1)** | 0.970 $\pm$ 0.007 (3) | 0.972 $\pm$ 0.008 (2) |
| ilpd | 0.657 $\pm$ 0.042 (2) | 0.652 $\pm$ 0.047 (3) | **0.663 $\pm$ 0.047 (1)** |
| tokyo1 | **0.921 $\pm$ 0.004 (1)** | 0.920 $\pm$ 0.010 (3) | 0.921 $\pm$ 0.008 (2) |
| qsar-biodeg | 0.854 $\pm$ 0.022 (3) | 0.855 $\pm$ 0.018 (2) | **0.845 $\pm$ 0.018 (1)** |
| ozone-level-8hr | **0.726 $\pm$ 0.020 (1)** | 0.710 $\pm$ 0.024 (3) | 0.707 $\pm$ 0.031 (2) |
| madelon | **0.803 $\pm$ 0.010 (1)** | 0.773 $\pm$ 0.009 (2) | 0.747 $\pm$ 0.009 (3) |
| Bioresponse | 0.794 $\pm$ 0.008 (2) | **0.795 $\pm$ 0.012 (1)** | 0.792 $\pm$ 0.010 (3) |
| wilt | **0.936 $\pm$ 0.015 (1)** | 0.932 $\pm$ 0.014 (2) | 0.929 $\pm$ 0.012 (3) |
| churn | **0.914 $\pm$ 0.017 (1)** | 0.899 $\pm$ 0.010 (2) | 0.887 $\pm$ 0.015 (3) |
| phoneme | **0.846 $\pm$ 0.008 (1)** | 0.829 $\pm$ 0.002 (2) | 0.828 $\pm$ 0.010 (3) |
| SpeedDating | 0.723 $\pm$ 0.013 (3) | **0.725 $\pm$ 0.012 (1)** | 0.725 $\pm$ 0.012 (2) |
| PhishingWebsites | **0.969 $\pm$ 0.006 (1)** | 0.968 $\pm$ 0.006 (2) | 0.967 $\pm$ 0.006 (3) |
| Amazon_employee_access | **0.665 $\pm$ 0.009 (1)** | 0.663 $\pm$ 0.010 (3) | 0.664 $\pm$ 0.016 (2) |
| nomao | **0.958 $\pm$ 0.002 (1)** | 0.956 $\pm$ 0.002 (2) | 0.954 $\pm$ 0.003 (3) |
| adult | 0.790 $\pm$ 0.006 (2) | **0.793 $\pm$ 0.005 (1)** | 0.790 $\pm$ 0.006 (3) |
| numerai28.6 | 0.519 $\pm$ 0.003 (2) | **0.520 $\pm$ 0.004 (1)** | 0.519 $\pm$ 0.003 (3) |
| Normalized Mean $\uparrow$ | **0.791 (1)** | 0.419 (2) | 0.221 (3) |
| Mean Reciprocal Rank (MRR) $\uparrow$ | **0.825 (1)** | 0.553 (2) | 0.456 (3) |

Table 11: **Ablation Study Weighting.** We report the test macro F1-Score (mean $\pm$ stdev for a 5-fold CV) with optimized parameters (complete grids, 250 trials) and the ranking of each approach in parentheses. The datasets are sorted based on the data size.

|  | Leaf Weights | Estimator Weights |
|---|---|---|
| dresses-sales | **0.612 $\pm$ 0.049 (1)** | 0.605 $\pm$ 0.050 (2) |
| climate-simulation-crashes | **0.853 $\pm$ 0.070 (1)** | 0.801 $\pm$ 0.040 (2) |
| cylinder-bands | **0.819 $\pm$ 0.032 (1)** | 0.787 $\pm$ 0.051 (2) |
| wdbc | **0.975 $\pm$ 0.010 (1)** | 0.970 $\pm$ 0.009 (2) |
| ilpd | **0.657 $\pm$ 0.042 (1)** | 0.612 $\pm$ 0.064 (2) |
| tokyo1 | 0.921 $\pm$ 0.004 (2) | **0.922 $\pm$ 0.016 (1)** |
| qsar-biodeg | **0.854 $\pm$ 0.022 (1)** | 0.850 $\pm$ 0.019 (2) |
| ozone-level-8hr | **0.726 $\pm$ 0.020 (1)** | 0.711 $\pm$ 0.020 (2) |
| madelon | **0.803 $\pm$ 0.010 (1)** | 0.606 $\pm$ 0.028 (2) |
| Bioresponse | **0.794 $\pm$ 0.008 (1)** | 0.784 $\pm$ 0.010 (2) |
| wilt | **0.936 $\pm$ 0.015 (1)** | 0.930 $\pm$ 0.019 (2) |
| churn | **0.914 $\pm$ 0.017 (1)** | 0.873 $\pm$ 0.018 (2) |
| phoneme | **0.846 $\pm$ 0.008 (1)** | 0.845 $\pm$ 0.011 (2) |
| SpeedDating | 0.723 $\pm$ 0.013 (2) | **0.728 $\pm$ 0.009 (1)** |
| PhishingWebsites | **0.969 $\pm$ 0.006 (1)** | 0.965 $\pm$ 0.006 (2) |
| Amazon_employee_access | 0.665 $\pm$ 0.009 (2) | **0.675 $\pm$ 0.008 (1)** |
| nomao | **0.958 $\pm$ 0.002 (1)** | 0.954 $\pm$ 0.002 (2) |
| adult | **0.790 $\pm$ 0.006 (1)** | 0.790 $\pm$ 0.005 (2) |
| numerai28.6 | **0.519 $\pm$ 0.003 (1)** | 0.519 $\pm$ 0.003 (2) |
| Normalized Mean $\uparrow$ | **0.824 (1)** | 0.177 (2) |
| Mean Reciprocal Rank (MRR) $\uparrow$ | **0.921 (1)** | 0.579 (2) |

Table 12: **Pairwise Confusion Matrix PhishingWebsites** We compare the predictions of each approach with the predictions of a CART DT. It becomes evident, that a simple model is not sufficient to solve the task well, as CART makes more than twice as many mistakes as state-of-the-art models.

|  | Correct DT | Incorrect DT | Total |
|---|---|---|---|
| Correct GRANDE | 2012 | 128 | **2140** |
| Incorrect GRANDE | 15 | 56 | **71** |
| Total | **2027** | **184** |  |

|  | Correct DT | Incorrect DT | Total |
|---|---|---|---|
| Correct XGBoost | 2018 | 115 | **2133** |
| Incorrect XGBoost | 9 | 69 | **78** |
| Total | **2027** | **184** |  |

|  | Correct DT | Incorrect DT | Total |
|---|---|---|---|
| Correct CatBoost | 2019 | 105 | **2124** |
| Incorrect CatBoost | 8 | 79 | **87** |
| Total | **2027** | **184** |  |

|  | Correct DT | Incorrect DT | Total |
|---|---|---|---|
| Correct NODE | 2012 | 112 | **2124** |
| Incorrect NODE | 15 | 72 | **87** |
| Total | **2027** | **184** |  |

## C  ADDITIONAL BENCHMARKS

In addition to the representative evaluation in the main paper, we provide an extended comparison with additional benchmarks, including tree ensemble methods Random Forest (Breiman, 2001) and ExtraTrees (Geurts et al., 2006), as well as SAINT Somepalli et al. (2021) as an additional DL benchmark, which is the superior gradient-based method according to Borisov et al. (2022). In the following, we present results using default hyperparameters for all methods (Table 15) and optimized hyperparameters (Table 13 and Table 14). The results for HPO are presented twofold. In Table 13, we report the results of an extensive search comprising 250 trials. Due to the very long runtime of SAINT, we were not able to optimize 250 trials for SAINT. Therefore, we present results from an

Table 13: **Extended Performance Comparison (HPO 250 trials).** We report the test macro F1-score (mean for a 5-fold CV) with optimized parameters based on a HPO with 250 trials. The datasets are sorted based on the data size.

| | GRANDE | XGB | CatBoost | NODE | ExtraTree | RandomForest |
|---|---|---|---|---|---|---|
| dresses-sales | **0.612 (1)** | 0.581 (4) | 0.588 (3) | 0.564 (5) | 0.589 (2) | 0.559 (6) |
| climate-model-simulation | **0.853 (1)** | 0.763 (4) | 0.778 (3) | 0.802 (2) | 0.754 (5) | 0.751 (6) |
| cylinder-bands | **0.819 (1)** | 0.773 (4) | 0.801 (2) | 0.754 (6) | 0.762 (5) | 0.782 (3) |
| wdbc | **0.975 (1)** | 0.953 (4) | 0.963 (3) | 0.966 (2) | 0.951 (5) | 0.949 (6) |
| ilpd | **0.657 (1)** | 0.632 (5) | 0.643 (3) | 0.526 (6) | 0.645 (2) | 0.640 (4) |
| tokyo1 | 0.921 (3) | 0.915 (5) | **0.927 (1)** | 0.921 (2) | 0.916 (4) | 0.909 (6) |
| qsar-biodeg | 0.854 (2) | 0.853 (3) | 0.844 (5) | 0.836 (6) | 0.851 (4) | **0.855 (1)** |
| ozone-level-8hr | **0.726 (1)** | 0.688 (6) | 0.721 (2) | 0.703 (3) | 0.695 (5) | 0.697 (4) |
| madelon | 0.803 (3) | 0.833 (2) | **0.861 (1)** | 0.571 (6) | 0.766 (4) | 0.764 (5) |
| Bioresponse | 0.794 (3) | 0.799 (2) | **0.801 (1)** | 0.780 (6) | 0.794 (5) | 0.794 (4) |
| wilt | 0.936 (2) | 0.911 (6) | 0.919 (3) | **0.937 (1)** | 0.917 (5) | 0.918 (4) |
| churn | 0.914 (2) | 0.900 (3) | 0.869 (6) | **0.930 (1)** | 0.899 (4) | 0.899 (5) |
| phoneme | 0.846 (6) | 0.872 (2) | **0.876 (1)** | 0.862 (5) | 0.868 (4) | 0.868 (3) |
| SpeedDating | **0.723 (1)** | 0.704 (6) | 0.718 (2) | 0.707 (5) | 0.708 (3) | 0.707 (4) |
| PhishingWebsites | **0.969 (1)** | 0.968 (2) | 0.965 (4) | 0.968 (3) | 0.963 (5) | 0.963 (5) |
| Amazon_employee_access | 0.665 (2) | 0.621 (6) | **0.671 (1)** | 0.649 (3) | 0.642 (5) | 0.648 (4) |
| nomao | 0.958 (3) | **0.965 (1)** | 0.964 (2) | 0.956 (4) | 0.951 (6) | 0.955 (5) |
| adult | 0.790 (5) | **0.798 (1)** | 0.796 (2) | 0.794 (3) | 0.785 (6) | 0.790 (4) |
| numerai28.6 | **0.519 (1)** | 0.518 (5) | 0.519 (2) | 0.503 (6) | 0.519 (4) | 0.519 (3) |
| Normalized Mean ↑ | **0.817 (1)** | 0.518 (3) | 0.705 (2) | 0.382 (6) | 0.385 (5) | 0.404 (4) |
| Mean Reciprocal Rank (MRR) ↑ | **0.668 (1)** | 0.365 (3) | 0.541 (2) | 0.352 (4) | 0.251 (6) | 0.275 (5) |

additional HPO consisting of 30 trials for each method (similar to McElfresh et al. (2023)) including SAINT (Table 13). Details on the HPO can be found in Appendix E. The implementation for SAINT, similar to NODE, was adopted from Borisov et al. (2022).

In all experiments, GRANDE demonstrated superior results over existing methods, achieving the highest normalized accuracy, mean reciprocal rank (MRR) and number of wins in almost all scenarios. In line with the experiments from the paper, CatBoost achieved the second-best and XGBoost the third-best results. The only exception is using default parameters, where XGBoost achieved a slightly higher normalized mean, but GRANDE still an significant higher MRR. While SAINT provided competitive results for some datasets (1 wins with HPO), it also achieved a very performance in some cases (e.g. for the *madelon* dataset, the prediction of SAINT was only predicting the majority class). Similarly, ExtraTree and RandomForest, especially with optimized hyperparameters achieved competitive results, but were almost consistently outperformed by gradient-boosting methods and GRANDE.

Table 14: **Extended Performance Comparison (HPO 30 trials).** We report the test macro F1-score (mean for a 5-fold CV) with optimized parameters (30 trials). The datasets are sorted based on the data size.

| | GRANDE | XGB | CatBoost | NODE | ExtraTree | RandomForest | SAINT |
|---|---|---|---|---|---|---|---|
| dresses-sales | **0.596 (1)** | 0.567 (5) | 0.592 (2) | 0.564 (6) | 0.583 (3) | 0.570 (4) | 0.516 (7) |
| climate-model-simulation-crashes | **0.841 (1)** | 0.764 (6) | 0.782 (5) | 0.802 (3) | 0.802 (2) | 0.793 (4) | 0.744 (7) |
| cylinder-bands | **0.799 (1)** | 0.769 (4) | 0.787 (3) | 0.754 (6) | 0.766 (5) | 0.753 (7) | 0.791 (2) |
| wdbc | 0.966 (2) | 0.959 (4) | 0.964 (3) | **0.966 (1)** | 0.955 (5) | 0.951 (7) | 0.952 (6) |
| ilpd | **0.656 (1)** | 0.625 (5) | 0.650 (2) | 0.526 (6) | 0.650 (3) | 0.642 (4) | 0.425 (7) |
| tokyo1 | **0.933 (1)** | 0.919 (6) | 0.925 (3) | 0.921 (4) | 0.929 (2) | 0.920 (5) | 0.915 (7) |
| qsar-biodeg | **0.857 (1)** | 0.841 (6) | 0.850 (5) | 0.836 (7) | 0.850 (4) | 0.855 (2) | 0.850 (3) |
| ozone-level-8hr | 0.716 (2) | 0.676 (6) | **0.736 (1)** | 0.703 (3) | 0.695 (5) | 0.697 (4) | 0.659 (7) |
| madelon | 0.787 (3) | 0.833 (2) | **0.861 (1)** | 0.571 (6) | 0.656 (5) | 0.774 (4) | 0.333 (7) |
| Bioresponse | 0.794 (5) | **0.799 (1)** | 0.797 (2) | 0.780 (6) | 0.796 (3) | 0.795 (4) | 0.780 (6) |
| wilt | 0.934 (3) | 0.913 (6) | 0.919 (4) | 0.937 (2) | **0.938 (1)** | 0.916 (5) | 0.574 (7) |
| churn | 0.914 (2) | 0.899 (4) | 0.869 (7) | **0.930 (1)** | 0.903 (3) | 0.899 (5) | 0.873 (6) |
| phoneme | 0.855 (6) | 0.870 (2) | **0.881 (1)** | 0.862 (4) | 0.852 (7) | 0.868 (3) | 0.858 (5) |
| SpeedDating | **0.728 (1)** | 0.704 (6) | 0.715 (2) | 0.707 (5) | 0.709 (4) | 0.709 (3) | 0.678 (7) |
| PhishingWebsites | 0.966 (4) | **0.968 (1)** | 0.964 (5) | 0.968 (2) | 0.953 (7) | 0.962 (6) | 0.966 (3) |
| Amazon_employee_access | 0.622 (6) | 0.620 (7) | 0.671 (2) | 0.649 (3) | 0.633 (5) | 0.647 (4) | **0.696 (1)** |
| nomao | 0.955 (5) | 0.963 (2) | **0.964 (1)** | 0.956 (3) | 0.944 (7) | 0.954 (6) | 0.956 (4) |
| adult | 0.790 (5) | **0.798 (1)** | 0.796 (2) | 0.794 (3) | 0.784 (6) | 0.790 (4) | 0.774 (7) |
| numerai28.6 | **0.521 (1)** | 0.518 (5) | 0.519 (3) | 0.503 (7) | 0.518 (4) | 0.520 (2) | 0.509 (6) |
| Normalized Mean ↑ | **0.804 (1)** | 0.609 (3) | 0.787 (2) | 0.510 (6) | 0.523 (5) | 0.585 (4) | 0.244 (7) |
| Mean Reciprocal Rank (MRR) ↑ | **0.597 (1)** | 0.368 (3) | 0.491 (2) | 0.341 (4) | 0.299 (5) | 0.257 (6) | 0.240 (7) |

Table 15: **Extended Performance Comparison (Default Parameters).** We report the test macro F1-score (mean for a 5-fold CV) with default parameters. The datasets are sorted based on the data size.

| | GRANDE | XGB | CatBoost | NODE | ExtraTree | RandomForest | SAINT |
|---|---|---|---|---|---|---|---|
| dresses-sales | **0.596 (1)** | 0.570 (3) | 0.573 (2) | 0.559 (4) | 0.545 (6) | 0.554 (5) | 0.422 (7) |
| climate-model-simulation-crashes | 0.758 (5) | 0.781 (2) | 0.781 (3) | 0.766 (4) | 0.728 (6) | 0.714 (7) | **0.850 (1)** |
| cylinder-bands | **0.813 (1)** | 0.770 (5) | 0.795 (2) | 0.696 (7) | 0.780 (4) | 0.787 (3) | 0.733 (6) |
| wdbc | 0.962 (3) | 0.966 (1) | 0.955 (4) | 0.964 (2) | 0.940 (7) | 0.942 (6) | 0.945 (5) |
| ilpd | **0.646 (1)** | 0.629 (3) | 0.643 (2) | 0.501 (6) | 0.582 (5) | 0.592 (4) | 0.416 (7) |
| tokyo1 | **0.922 (1)** | 0.917 (6) | 0.917 (5) | 0.921 (2) | 0.917 (3) | 0.917 (4) | 0.907 (7) |
| qsar-biodeg | **0.851 (1)** | 0.844 (2) | 0.843 (3) | 0.838 (4) | 0.836 (7) | 0.838 (5) | 0.836 (6) |
| ozone-level-8hr | **0.735 (1)** | 0.686 (3) | 0.702 (2) | 0.662 (4) | 0.601 (7) | 0.605 (6) | 0.651 (5) |
| madelon | 0.768 (5) | 0.811 (2) | **0.851 (1)** | 0.650 (6) | 0.773 (4) | 0.773 (3) | 0.333 (7) |
| Bioresponse | 0.789 (4) | 0.789 (5) | 0.792 (3) | 0.786 (6) | 0.799 (2) | **0.801 (1)** | 0.786 (6) |
| wilt | **0.933 (1)** | 0.903 (5) | 0.898 (6) | 0.904 (4) | 0.916 (2) | 0.911 (3) | 0.486 (7) |
| churn | 0.896 (3) | 0.897 (2) | 0.862 (7) | **0.925 (1)** | 0.892 (5) | 0.893 (4) | 0.867 (6) |
| phoneme | 0.860 (6) | 0.864 (3) | 0.861 (5) | 0.842 (7) | 0.875 (2) | **0.880 (1)** | 0.862 (4) |
| SpeedDating | **0.725 (1)** | 0.686 (4) | 0.693 (3) | 0.703 (2) | 0.642 (6) | 0.640 (7) | 0.643 (5) |
| PhishingWebsites | **0.969 (1)** | 0.969 (2) | 0.963 (6) | 0.961 (7) | 0.968 (3) | 0.968 (3) | 0.964 (5) |
| Amazon_employee_access | 0.602 (7) | 0.608 (6) | 0.652 (4) | 0.621 (5) | 0.662 (3) | **0.682 (1)** | 0.682 (2) |
| nomao | 0.955 (6) | **0.965 (1)** | 0.962 (2) | 0.955 (7) | 0.956 (4) | 0.961 (3) | 0.956 (5) |
| adult | 0.785 (5) | 0.796 (2) | 0.796 (3) | **0.799 (1)** | 0.784 (6) | 0.793 (4) | 0.768 (7) |
| numerai28.6 | 0.503 (7) | 0.516 (2) | **0.519 (1)** | 0.506 (6) | 0.514 (3) | 0.512 (4) | 0.509 (5) |
| Normalized Mean ↑ | 0.673 (2) | **0.691 (1)** | 0.666 (3) | 0.460 (6) | 0.518 (5) | 0.596 (4) | 0.228 (7) |
| Mean Reciprocal Rank (MRR) ↑ | **0.586 (1)** | 0.422 (2) | 0.397 (3) | 0.326 (5) | 0.267 (6) | 0.365 (4) | 0.235 (7) |

# D DISCUSSION WEIGHTING STATISTICS

Within our paper, we showed in an ablation study that our instance-wise weighting has a positive impact on the performance of GRANDE. In addition, we showcased a real-world scenario where the weighting significantly increases local interpretability by enabling the model to learn representations for simple and complex rules. In the following, we provide a more detailed evaluation of the instance-wise weighting, including statistics on the highest weighted estimators and the weight distributions.

The following statistics were obtained by calculating the post-softmax weights for each sample and averaging the values over all samples. In addition, we calculated the same statistics for the top 5% of samples with the highest weights. This provides additional insights since we agrue that unique local interactions might exist only for a subset of the samples.

Table 16: **Weighting Statistics Estimator**

| | Number Internal Nodes | Number Leaf Nodes | Percentage Highest Estimator | Percentage Highest Estimator Top 5% | Count Highest Estimator | Count Highest Estimator Top 5% |
|---|---|---|---|---|---|---|
| dresses-sales | 1.12 | 2.12 | 0.470 | 0.667 | 10.20 | 2.40 |
| climate-model-simulation | 3.12 | 4.12 | 0.241 | 0.467 | 15.80 | 4.00 |
| cylinder-bands | 5.44 | 6.44 | 0.124 | 0.400 | 31.20 | 4.00 |
| wdbc | 1.16 | 2.16 | 0.587 | 0.733 | 4.20 | 1.60 |
| ilpd | 3.96 | 4.96 | 0.319 | 0.600 | 15.00 | 3.40 |
| tokyo1 | 3.20 | 4.20 | 0.273 | 0.780 | 19.80 | 2.00 |
| qsar-biodeg | 2.80 | 3.80 | 0.191 | 0.745 | 29.80 | 2.60 |
| ozone-level-8hr | 2.00 | 3.00 | 0.198 | 0.662 | 27.80 | 5.40 |
| madelon | 3.92 | 4.92 | 0.322 | 0.741 | 19.60 | 3.20 |
| Bioresponse | 3.96 | 4.96 | 0.099 | 0.311 | 82.00 | 11.80 |
| wilt | 1.76 | 2.76 | 0.274 | 0.604 | 19.20 | 5.00 |
| churn | 2.28 | 3.28 | 0.111 | 0.408 | 72.00 | 11.40 |
| phoneme | 3.28 | 4.28 | 0.163 | 0.636 | 40.80 | 4.40 |
| SpeedDating | 6.96 | 7.96 | 0.168 | 0.640 | 90.20 | 5.00 |
| PhishingWebsites | 5.60 | 6.60 | 0.178 | 0.594 | 116.80 | 12.20 |
| Amazon_employee_access | 0.24 | 1.24 | 0.882 | 0.999 | 2.80 | 1.20 |
| nomao | 3.20 | 4.20 | 0.113 | 0.260 | 218.20 | 15.80 |
| adult | 0.36 | 1.36 | 0.905 | 0.912 | 5.20 | 1.60 |
| numerai28.6 | 1.08 | 2.08 | 0.559 | 0.824 | 19.60 | 4.40 |
| Mean | 2.92 | 3.92 | 0.325 | 0.631 | 44.22 | 5.35 |

We can observe that on average, the highest weighted estimator comprises a moderate number of ≈3 internal and ≈4 leaf nodes, allowing an easy interpretation (see Table 16). Furthermore, on average, the highest weighted estimator is the same for ≈33% of the data instances, with a total of ≈44 different estimators having the highest weight for at least one instance. If we further inspect the top 5% of instances with the highest weight for a single estimator, we can additionally observe that the highest weighted estimator is the same for ≈63% of these instances and only 5 different estimators have the highest weight for at least one instance. This suggests the presence of a limited number of local experts in most datasets, which have high weights for specific instances where local rules are applicable.

In addition, Table 17 summarizes the skewness as measure of the asymmetry of a distribution, and kurtosis as measure of the tailedness of a distribution, for all datasets. In general, it stands out that both values increase significantly when considering only the top 5% of samples with the highest weight for a single estimator. This again indicates the presence of local expert estimators for a subset of the data, where unique local interactions were identified. In addition, there are major differences in the values depending on the datasets, which we will discuss more detailed in the following:

**Skewness**   In total, for 16/19 datasets the weights have a skewed (10) or very skewed (6) distribution, i.e., are long-tailed (see Table 18). Generally, left-skewed distributions suggest a few estimators with very low weights, thereby contributing significantly less and resulting in a more compact ensemble. However, right-skewed distributions are more desired as they indicate a small number of trees having high weights (which we consider as local experts), while the majority of estimators have small weights. Considering the top 5% of samples with the highest weights of a single estimator, we can see that the number of very skewed distributions increases from 6 to 13 (see Table 19), indicating samples that can be assigned to a class more easily based on a small number of trees (similar to the example in the case study). In general, we are most interested in right-skewed distributions as this indicates a small number of estimators with high weights, which we can consider as local experts.

**(Excess) Kurtosis**   We can observe that 11/19 distributions are leptokurtic, i.e., have heavy tails, with 4 considered as very leptokurtic (see Table 20). In general, distributions with heavy tails are interesting, as this indicates outliers (estimators with significantly higher / lower weights). Trees comprising significantly higher weights can be considered as local experts that have learned unique local interactions. Again, when considering the top 5% of samples with the highest weights of a single estimator, the number of leptokurtic distributions increases from 11 to 15 with the number of

Table 17: **Weighting Statistics Distribution.** We provide the skewness, as measure of the asymmetry of a distribution, and kurtosis, as measure of the tailedness of a distribution, for all datasets. The measures are split in an average over all samples and an average over the top 5% of the samples with the highest maximum weight.

|  | Skewness | Skewness Top 5% | Kurtosis | Kurtosis Top 5% |
|---|---|---|---|---|
| dresses-sales | 0.946 | 1.187 | 0.617 | 1.597 |
| climate-model-simulation | 1.259 | 1.909 | 0.742 | 3.095 |
| cylinder-bands | 0.853 | 1.199 | -0.562 | 0.230 |
| wdbc | 0.082 | 0.251 | -1.084 | -0.980 |
| ilpd | 0.180 | 0.385 | -0.649 | -0.528 |
| tokyo1 | 0.975 | 1.839 | 1.322 | 4.919 |
| qsar-biodeg | 0.875 | 1.194 | 0.200 | 1.302 |
| ozone-level-8hr | 0.752 | 1.462 | -0.048 | 1.972 |
| madelon | 1.901 | 2.208 | 2.797 | 4.465 |
| Bioresponse | 0.777 | 1.139 | -0.496 | 0.317 |
| wilt | 0.508 | 1.150 | -0.499 | 1.100 |
| churn | 2.369 | 3.488 | 5.912 | 13.419 |
| phoneme | 0.563 | 0.789 | -0.447 | 0.119 |
| SpeedDating | 1.069 | 1.180 | 1.148 | 1.627 |
| PhishingWebsites | 1.263 | 1.978 | 1.136 | 4.343 |
| Amazon_employee_access | -0.029 | -0.119 | 3.348 | 3.081 |
| nomao | 2.411 | 4.190 | 6.920 | 24.791 |
| adult | 0.621 | 0.679 | 3.345 | 3.280 |
| numerai28.6 | -0.297 | -0.269 | 0.833 | 0.829 |
| Mean | 0.899 | 1.360 | 1.291 | 3.630 |

Table 18: **Skewness**

|  | Very skewed $(-\infty, -1.0)$ or $(1.0, \infty)$ | Skewed $[-1.0, -0.5)$ or $(0.5, 1.0]$ | Slight skew $[-0.5, 0.5]$ | **Sum** |
|---|---|---|---|---|
| Left skew | 0 | 1 | 1 | **2** |
| Right skew | 6 | 9 | 2 | **17** |
| **Sum** | **6** | **10** | **3** |  |

very leptokurtic distributions increasing from 4 to 8. Again, this indicates that there exists samples that can be assigned to a class more easily with simple rules (from local experts) in many datasets.

Overall, the additional statistics are in line with the discussion in the paper. For several datasets, the distribution of weight is long- and/or heavy-tailed, indicating the presence of local expert estimators. However, the benefit of learning local expert estimators is dataset-dependent and not feasible in all scenarios. This is also confirmed by our results, as there are datasets where the distribution is symmetric and mesokurtic. This is valid for instance for numerai28.6, which is a very complex dataset without many simple rules (as indicated by the poor performance of all methods). In addition, it should be noted that a more comprehensive analysis is necessary to confirm these additional findings.

Table 19: **Skewness Top 5%**

|  | Very skewed $(-\infty, -1.0)$ or $(1.0, \infty)$ | Skewed $[-1.0, -0.5)$ or $(0.5, 1.0]$ | Slight skew $[-0.5, 0.5]$ | Sum |
|---|---|---|---|---|
| Left skew | 0 | 1 | 1 | **2** |
| Right skew | 13 | 1 | 3 | **17** |
| **Sum** | **13** | **2** | **4** |  |

Table 20: **Kurtosis**

| Excess Kurtosis | Count | Count Top 5% |
|---|---|---|
| Very leptokurtic $(3.0, \infty)$ | 4 | 8 |
| Moderately leptokurtic $(0.5, 3.0]$ | 7 | 6 |
| Mesokurtic $[-0.5, 0.5]$ | 5 | 3 |
| Platykurtic $(-\infty, -0.5)$ | 3 | 2 |

# E    HYPERPARAMETERS

We optimized the hyperparameters using Optuna (Akiba et al., 2019) with 250 trials and selected the search space and default parameters for related work in accordance with Borisov et al. (2022). The best parameters were selected based on a 5x2 cross-validation as suggested by Raschka (2018) where the test data of each fold was held out of the HPO to get unbiased results. To deal with class imbalance, we further included class weights. In line with Borisov et al. (2022), we did not tune the number of estimators for XGBoost and CatBoost, but used early stopping. To validate our approach of not tuning the estimators, we conducted an additional HPO with tuning the estimators for XGBoost and CatBoost (see Table 21 and Table 22) and show that this results in similar (slightly worse) results.

While using 250 trials can be considered as a very exhaustive search, we include additional results for a HPO with only 30 trials, similar to McElfresh et al. (2023). Based on the results in Table 23, we can verify that even with a small number of trials, GRANDE achieves strong results. Yet, we can also observe that searching for more trials is especially beneficial for GRANDE.

Furthermore, GRANDE has a total of 11 hyperparameters that were optimized during the HPO. In contrast, for XGBoost and CatBoost we only optimized 4 and 3 hyperparameters, respectively. The choice to optimize only a small number of hyperparameters was in accordance with Borisov et al. (2022) based on the assumption that an exhaustive search on the most relevant parameters is beneficial for the performance. To account for this, we performed an additional HPO for GRANDE, where we similarly only optimized 4 parameters (one overall learning_rate, focal_factor, cosine_decay_steps and dropout). The results are displayed in Table 24 and show that GRANDE still achieves SOTA results. However, it becomes evident, that GRANDE benefits from using a larger grid, especially since using different learning rates for the different components (split values, split indices, leaf probabilities and leaf weights) has a positive impact on the performance.

For GRANDE, we used a batch size of 64 and early stopping after 25 epochs. Similar to NODE Popov et al. (2019), GRANDE uses an Adam optimizer with stochastic weight averaging over 5 checkpoints (Izmailov et al., 2018) and a learning rate schedule that uses a cosine decay with optional warmup (Loshchilov & Hutter, 2016). Furthermore, GRANDE allows using a focal factor (Lin et al., 2017), similar to GradTree Marton et al. (2023). In the supplementary material, we provide the notebook used for the optimization along with the search space for each approach.

Table 21: **Comparison XGB HPO (250 Trials).** We compare a HPO for XGBoost with and without tuning the number of estimators based on 250 trials. Tuning the estimators does not improve the performance, and using early stopping in addition to setting the number of estimators to a high number is sufficient.

| | Without tuning estimators | With tuning estimators | Difference |
|---|---|---|---|
| dresses-sales | 0.5755 | **0.5813** | -0.0058 |
| climate-model-simulation-crashes | **0.7932** | 0.7626 | 0.0306 |
| cylinder-bands | 0.7670 | **0.7734** | -0.0064 |
| wdbc | **0.9568** | 0.9531 | 0.0037 |
| ilpd | 0.6316 | **0.6321** | -0.0005 |
| tokyo1 | **0.9147** | 0.9147 | 0.0000 |
| qsar-biodeg | 0.8486 | **0.8526** | -0.0039 |
| ozone-level-8hr | **0.7044** | 0.6885 | 0.0159 |
| madelon | 0.8226 | **0.8334** | -0.0108 |
| Bioresponse | 0.7924 | **0.7988** | -0.0064 |
| wilt | 0.9102 | **0.9114** | -0.0012 |
| churn | **0.9003** | 0.8998 | 0.0005 |
| phoneme | **0.8725** | 0.8718 | 0.0007 |
| SpeedDating | **0.7071** | 0.7043 | 0.0028 |
| PhishingWebsites | 0.9674 | **0.9683** | -0.0009 |
| Amazon_employee_access | **0.6225** | 0.6209 | 0.0016 |
| nomao | 0.9642 | **0.9650** | -0.0008 |
| adult | 0.7977 | **0.7980** | -0.0003 |
| numerai28.6 | **0.5187** | 0.5180 | 0.0007 |
| Mean ↑ | **0.7930** | 0.7920 | 0.0010 |

Table 22: **Comparison CatBoost HPO (250 Trials).** We compare a HPO for CatBoost with and without tuning the number of estimators based on 250 trials. Tuning the estimators does not improve the performance, and using early stopping in addition to setting the number of estimators to a high number is sufficient.

| | Without tuning estimators | With tuning estimators | Difference |
|---|---|---|---|
| dresses-sales | 0.5611 | **0.5880** | -0.0270 |
| climate-model-simulation-crashes | **0.7902** | 0.7775 | 0.0127 |
| cylinder-bands | **0.8080** | 0.8014 | 0.0066 |
| wdbc | 0.9551 | **0.9625** | -0.0074 |
| ilpd | 0.6379 | **0.6428** | -0.0049 |
| tokyo1 | 0.9249 | **0.9274** | -0.0025 |
| qsar-biodeg | **0.8467** | 0.8444 | 0.0023 |
| ozone-level-8hr | **0.7330** | 0.7206 | 0.0124 |
| madelon | **0.8650** | 0.8607 | 0.0042 |
| Bioresponse | 0.7992 | **0.8013** | -0.0021 |
| wilt | **0.9227** | 0.9187 | 0.0040 |
| churn | **0.8734** | 0.8693 | 0.0042 |
| phoneme | **0.8828** | 0.8763 | 0.0065 |
| SpeedDating | 0.7182 | **0.7184** | -0.0001 |
| PhishingWebsites | **0.9657** | 0.9650 | 0.0007 |
| Amazon_employee_access | 0.6704 | **0.6709** | -0.0004 |
| nomao | 0.9633 | **0.9644** | -0.0011 |
| adult | 0.7957 | **0.7960** | -0.0003 |
| numerai28.6 | 0.5172 | **0.5193** | -0.0021 |
| Mean ↑ | **0.8016** | 0.8013 | 0.0003 |

Table 23: **HPO 30 Trials Performance Comparison.** We report the test macro F1-score (mean $\pm$ stdev over 10 trials) based on a reduced HPO with only 30 trials. The datasets are sorted based on the data size.

|  | GRANDE | XGB | CatBoost | NODE |
|---|---|---|---|---|
| dresses-sales | **0.596 ± 0.047 (1)** | 0.567 ± 0.055 (3) | 0.592 ± 0.046 (2) | 0.564 ± 0.051 (4) |
| climate-model-simulation | **0.841 ± 0.058 (1)** | 0.764 ± 0.073 (4) | 0.782 ± 0.057 (3) | 0.802 ± 0.035 (2) |
| cylinder-bands | **0.799 ± 0.037 (1)** | 0.769 ± 0.040 (3) | 0.787 ± 0.044 (2) | 0.754 ± 0.040 (4) |
| wdbc | 0.966 ± 0.019 (2) | 0.959 ± 0.026 (4) | 0.964 ± 0.022 (3) | **0.966 ± 0.016 (1)** |
| ilpd | **0.656 ± 0.044 (1)** | 0.625 ± 0.052 (3) | 0.650 ± 0.059 (2) | 0.526 ± 0.069 (4) |
| tokyo1 | **0.933 ± 0.011 (1)** | 0.919 ± 0.012 (4) | 0.925 ± 0.011 (2) | 0.921 ± 0.010 (3) |
| qsar-biodeg | **0.857 ± 0.025 (1)** | 0.841 ± 0.015 (3) | 0.850 ± 0.030 (2) | 0.836 ± 0.028 (4) |
| ozone-level-8hr | 0.716 ± 0.007 (2) | 0.676 ± 0.028 (4) | **0.736 ± 0.027 (1)** | 0.703 ± 0.029 (3) |
| madelon | 0.787 ± 0.014 (3) | 0.833 ± 0.015 (2) | **0.861 ± 0.012 (1)** | 0.571 ± 0.022 (4) |
| Bioresponse | 0.794 ± 0.010 (3) | **0.799 ± 0.011 (1)** | 0.797 ± 0.008 (2) | 0.780 ± 0.011 (4) |
| wilt | 0.934 ± 0.01 (2) | 0.913 ± 0.011 (4) | 0.919 ± 0.009 (3) | **0.937 ± 0.017 (1)** |
| churn | 0.914 ± 0.013 (2) | 0.899 ± 0.020 (3) | 0.869 ± 0.014 (4) | **0.930 ± 0.011 (1)** |
| phoneme | 0.855 ± 0.008 (4) | 0.870 ± 0.008 (2) | **0.881 ± 0.007 (1)** | 0.862 ± 0.013 (3) |
| SpeedDating | **0.728 ± 0.007 (1)** | 0.704 ± 0.012 (4) | 0.715 ± 0.015 (2) | 0.707 ± 0.015 (3) |
| PhishingWebsites | 0.966 ± 0.004 (3) | **0.968 ± 0.008 (1)** | 0.964 ± 0.004 (4) | 0.968 ± 0.006 (2) |
| Amazon_employee_access | 0.622 ± 0.028 (3) | 0.620 ± 0.008 (4) | **0.671 ± 0.013 (1)** | 0.649 ± 0.009 (2) |
| nomao | 0.955 ± 0.001 (3) | 0.963 ± 0.004 (2) | **0.964 ± 0.004 (1)** | 0.956 ± 0.001 (3) |
| adult | 0.790 ± 0.006 (4) | **0.798 ± 0.004 (1)** | 0.796 ± 0.005 (2) | 0.794 ± 0.004 (3) |
| numerai28.6 | **0.521 ± 0.004 (1)** | 0.518 ± 0.002 (2) | 0.519 ± 0.002 (2) | 0.503 ± 0.010 (4) |
| Normalized Mean ↑ | **0.694 (1)** | 0.430 (3) | 0.676 (2) | 0.336 (4) |
| Mean Reciprocal Rank (MRR) ↑ | **0.636 (1)** | 0.434 (3) | 0.579 (2) | 0.434 (3) |

Table 24: **HPO Reduced Grid Performance Comparison (250 Trials).** We report the test macro F1-score (mean $\pm$ stdev over 10 trials) based on a HPO with a reduced parameter grid for GRANDE (only one overall learning_rate, focal_factor, cosine_decay_steps and dropout) and 250 trials for all methods. The datasets are sorted based on the data size.

|  | GRANDE | XGB | CatBoost | NODE |
|---|---|---|---|---|
| dresses-sales | **0.590 ± 0.042 (1)** | 0.581 ± 0.059 (3) | 0.588 ± 0.036 (2) | 0.564 ± 0.051 (4) |
| climate-model-simulation | 0.800 ± 0.055 (2) | 0.763 ± 0.064 (4) | 0.778 ± 0.050 (3) | **0.802 ± 0.035 (1)** |
| cylinder-bands | 0.783 ± 0.040 (2) | 0.773 ± 0.042 (3) | **0.801 ± 0.043 (1)** | 0.754 ± 0.040 (4) |
| wdbc | 0.964 ± 0.007 (2) | 0.953 ± 0.030 (4) | 0.963 ± 0.023 (3) | **0.966 ± 0.016 (1)** |
| ilpd | **0.645 ± 0.030 (1)** | 0.632 ± 0.043 (3) | 0.643 ± 0.053 (2) | 0.526 ± 0.069 (4) |
| tokyo1 | **0.928 ± 0.017 (1)** | 0.915 ± 0.011 (4) | 0.927 ± 0.013 (2) | 0.921 ± 0.010 (3) |
| qsar-biodeg | **0.859 ± 0.024 (1)** | 0.853 ± 0.020 (2) | 0.844 ± 0.023 (3) | 0.836 ± 0.028 (4) |
| ozone-level-8hr | **0.733 ± 0.016 (1)** | 0.688 ± 0.021 (4) | 0.721 ± 0.027 (2) | 0.703 ± 0.029 (3) |
| madelon | 0.809 ± 0.010 (3) | 0.833 ± 0.018 (2) | **0.861 ± 0.012 (1)** | 0.571 ± 0.022 (4) |
| Bioresponse | 0.783 ± 0.009 (3) | 0.799 ± 0.011 (2) | **0.801 ± 0.014 (1)** | 0.780 ± 0.011 (4) |
| wilt | 0.921 ± 0.014 (2) | 0.911 ± 0.010 (4) | 0.919 ± 0.007 (3) | **0.937 ± 0.017 (1)** |
| churn | 0.888 ± 0.017 (3) | 0.900 ± 0.017 (2) | 0.869 ± 0.021 (4) | **0.930 ± 0.011 (1)** |
| phoneme | 0.859 ± 0.010 (4) | 0.872 ± 0.007 (2) | **0.876 ± 0.005 (1)** | 0.862 ± 0.013 (3) |
| SpeedDating | **0.725 ± 0.016 (1)** | 0.704 ± 0.015 (4) | 0.718 ± 0.014 (2) | 0.707 ± 0.015 (3) |
| PhishingWebsites | **0.970 ± 0.006 (1)** | 0.968 ± 0.006 (2) | 0.965 ± 0.003 (4) | 0.968 ± 0.006 (3) |
| Amazon_employee_access | 0.643 ± 0.024 (3) | 0.621 ± 0.008 (4) | **0.671 ± 0.011 (1)** | 0.649 ± 0.009 (2) |
| nomao | 0.956 ± 0.004 (4) | **0.965 ± 0.003 (1)** | 0.964 ± 0.002 (2) | 0.956 ± 0.001 (3) |
| adult | 0.791 ± 0.005 (4) | **0.798 ± 0.004 (1)** | 0.796 ± 0.004 (2) | 0.794 ± 0.004 (3) |
| numerai28.6 | 0.519 ± 0.005 (2) | 0.518 ± 0.001 (3) | **0.519 ± 0.002 (1)** | 0.503 ± 0.010 (4) |
| Normalized Mean ↑ | 0.659 ± 0.509 (2) | 0.488 ± 0.497 (3) | **0.721 ± 0.373 (1)** | 0.346 ± 0.561 (4) |
| Mean Reciprocal Rank (MRR) ↑ | **0.610 ± 0.531 (1)** | 0.425 ± 0.469 (4) | 0.596 ± 0.456 (2) | 0.452 ± 0.627 (3) |

Table 25: **Hyperparameters GRANDE (Part 1).** We report the hyperparameters for GRANDE based on the extensive HPO with 250 trials.

| | depth | n_estimators | lr_weights | lr_index | lr_values | lr_leaf |
|---|---|---|---|---|---|---|
| dresses-sales | 4 | 512 | 0.0015 | 0.0278 | 0.1966 | 0.0111 |
| climate-simulation-crashes | 4 | 2048 | 0.0007 | 0.0243 | 0.0156 | 0.0134 |
| cylinder-bands | 6 | 2048 | 0.0009 | 0.0084 | 0.0086 | 0.0474 |
| wdbc | 4 | 1024 | 0.0151 | 0.0140 | 0.1127 | 0.1758 |
| ilpd | 4 | 512 | 0.0007 | 0.0059 | 0.0532 | 0.0094 |
| tokyo1 | 6 | 1024 | 0.0029 | 0.1254 | 0.0056 | 0.0734 |
| qsar-biodeg | 6 | 2048 | 0.0595 | 0.0074 | 0.0263 | 0.0414 |
| ozone-level-8hr | 4 | 2048 | 0.0022 | 0.0465 | 0.0342 | 0.0503 |
| madelon | 4 | 2048 | 0.0003 | 0.0575 | 0.0177 | 0.0065 |
| Bioresponse | 6 | 2048 | 0.0304 | 0.0253 | 0.0073 | 0.0784 |
| wilt | 6 | 2048 | 0.0377 | 0.1471 | 0.0396 | 0.1718 |
| churn | 6 | 2048 | 0.0293 | 0.0716 | 0.0179 | 0.0225 |
| phoneme | 6 | 2048 | 0.0472 | 0.0166 | 0.0445 | 0.1107 |
| SpeedDating | 6 | 2048 | 0.0148 | 0.0130 | 0.0095 | 0.0647 |
| PhishingWebsites | 6 | 2048 | 0.0040 | 0.0118 | 0.0104 | 0.1850 |
| Amazon_employee_access | 6 | 2048 | 0.0036 | 0.0056 | 0.1959 | 0.1992 |
| nomao | 6 | 2048 | 0.0059 | 0.0224 | 0.0072 | 0.0402 |
| adult | 6 | 1024 | 0.0015 | 0.0087 | 0.0553 | 0.1482 |
| numerai28.6 | 4 | 512 | 0.0001 | 0.0737 | 0.0513 | 0.0371 |

Table 26: **Hyperparameters GRANDE (Part 2).** We report the hyperparameters for GRANDE based on the extensive HPO with 250 trials.

| | dropout | selected_variables | data_fraction | focal_factor | cosine_decay_steps |
|---|---|---|---|---|---|
| dresses-sales | 0.25 | 0.7996 | 0.9779 | 0 | 0.0 |
| climate-simulation-crashes | 0.00 | 0.6103 | 0.8956 | 0 | 1000.0 |
| cylinder-bands | 0.25 | 0.5309 | 0.8825 | 0 | 1000.0 |
| wdbc | 0.50 | 0.8941 | 0.8480 | 0 | 0.0 |
| ilpd | 0.50 | 0.6839 | 0.9315 | 3 | 1000.0 |
| tokyo1 | 0.50 | 0.5849 | 0.9009 | 0 | 1000.0 |
| qsar-biodeg | 0.00 | 0.5892 | 0.8098 | 0 | 0.0 |
| ozone-level-8hr | 0.25 | 0.7373 | 0.8531 | 0 | 1000.0 |
| madelon | 0.25 | 0.9865 | 0.9885 | 0 | 100.0 |
| Bioresponse | 0.50 | 0.5646 | 0.8398 | 0 | 0.0 |
| wilt | 0.25 | 0.9234 | 0.8299 | 0 | 0.0 |
| churn | 0.00 | 0.6920 | 0.8174 | 0 | 1000.0 |
| phoneme | 0.00 | 0.7665 | 0.8694 | 3 | 1000.0 |
| SpeedDating | 0.00 | 0.8746 | 0.8229 | 3 | 0.1 |
| PhishingWebsites | 0.00 | 0.9792 | 0.9588 | 0 | 0.1 |
| Amazon_employee_access | 0.50 | 0.9614 | 0.9196 | 3 | 0.0 |
| nomao | 0.00 | 0.8659 | 0.8136 | 0 | 100.0 |
| adult | 0.50 | 0.5149 | 0.8448 | 3 | 100.0 |
| numerai28.6 | 0.50 | 0.7355 | 0.8998 | 0 | 0.1 |

Table 27: **Hyperparameters XGBoost.** We report the hyperparameters for GRANDE based on the extensive HPO with 250 trials.

| | learning_rate | max_depth | reg_alpha | reg_lambda |
|---|---|---|---|---|
| dresses-sales | 0.1032 | 3 | 0.0000 | 0.0000 |
| climate-simulation-crashes | 0.0356 | 11 | 0.5605 | 0.0000 |
| cylinder-bands | 0.2172 | 11 | 0.0002 | 0.0057 |
| wdbc | 0.2640 | 2 | 0.0007 | 0.0000 |
| ilpd | 0.0251 | 4 | 0.3198 | 0.0000 |
| tokyo1 | 0.0293 | 3 | 0.2910 | 0.3194 |
| qsar-biodeg | 0.0965 | 5 | 0.0000 | 0.0000 |
| ozone-level-8hr | 0.0262 | 9 | 0.0000 | 0.6151 |
| madelon | 0.0259 | 6 | 0.0000 | 0.9635 |
| Bioresponse | 0.0468 | 5 | 0.9185 | 0.0000 |
| wilt | 0.1305 | 8 | 0.0000 | 0.0003 |
| churn | 0.0473 | 6 | 0.0000 | 0.3132 |
| phoneme | 0.0737 | 11 | 0.9459 | 0.2236 |
| SpeedDating | 0.0277 | 9 | 0.0000 | 0.9637 |
| PhishingWebsites | 0.1243 | 11 | 0.0017 | 0.3710 |
| Amazon_employee_access | 0.0758 | 11 | 0.9785 | 0.0042 |
| nomao | 0.1230 | 5 | 0.0000 | 0.0008 |
| adult | 0.0502 | 11 | 0.0000 | 0.7464 |
| numerai28.6 | 0.1179 | 2 | 0.0001 | 0.0262 |

Table 28: **Hyperparameters CatBoost.** We report the hyperparameters for GRANDE based on the extensive HPO with 250 trials.

| | learning_rate | max_depth | l2_leaf_reg |
|---|---|---|---|
| dresses-sales | 0.0675 | 3 | 19.8219 |
| climate-simulation-crashes | 0.0141 | 2 | 19.6955 |
| cylinder-bands | 0.0716 | 11 | 19.6932 |
| wdbc | 0.1339 | 3 | 0.7173 |
| ilpd | 0.0351 | 4 | 5.0922 |
| tokyo1 | 0.0228 | 5 | 0.5016 |
| qsar-biodeg | 0.0152 | 11 | 0.7771 |
| ozone-level-8hr | 0.0118 | 11 | 3.0447 |
| madelon | 0.0102 | 10 | 9.0338 |
| Bioresponse | 0.0195 | 11 | 8.1005 |
| wilt | 0.0192 | 11 | 1.1095 |
| churn | 0.0248 | 9 | 7.0362 |
| phoneme | 0.0564 | 11 | 0.6744 |
| SpeedDating | 0.0169 | 11 | 1.5494 |
| PhishingWebsites | 0.0239 | 8 | 1.6860 |
| Amazon_employee_access | 0.0123 | 11 | 1.6544 |
| nomao | 0.0392 | 8 | 2.6583 |
| adult | 0.1518 | 11 | 29.3098 |
| numerai28.6 | 0.0272 | 4 | 18.6675 |

Table 29: **Hyperparameters NODE.** We report the hyperparameters for GRANDE based on the grid suggested by the authors Popov et al. (2019).

| | num_layers | total_tree_count | tree_depth |
|---|---|---|---|
| dresses-sales | 2 | 2048 | 6 |
| climate-simulation-crashes | 2 | 1024 | 8 |
| cylinder-bands | 2 | 1024 | 8 |
| wdbc | 2 | 2048 | 6 |
| ilpd | 4 | 2048 | 8 |
| tokyo1 | 4 | 1024 | 8 |
| qsar-biodeg | 2 | 2048 | 6 |
| ozone-level-8hr | 4 | 1024 | 6 |
| madelon | 2 | 2048 | 6 |
| Bioresponse | 2 | 1024 | 8 |
| wilt | 4 | 1024 | 6 |
| churn | 2 | 1024 | 8 |
| phoneme | 4 | 2048 | 8 |
| SpeedDating | 4 | 1024 | 6 |
| PhishingWebsites | 2 | 1024 | 8 |
| Amazon_employee_access | 2 | 2048 | 6 |
| nomao | 2 | 2048 | 6 |
| adult | 4 | 1024 | 8 |
| numerai28.6 | 2 | 1024 | 8 |

