# OpenReview forum: "GRANDE: Gradient-Based Decision Tree Ensembles for Tabular Data"
_ICLR.cc/2024/Conference — ICLR 2024 poster_

### Official Review · Reviewer_SMN1 · 2023-10-23

**Soundness:** 3 good
**Presentation:** 3 good
**Contribution:** 2 fair
**Rating:** 6
**Confidence:** 5

**Summary:**

This paper presents GRANDE, a novel method for learning hard, axis-aligned decision tree ensembles using end-to-end gradient descent. The paper extends GradTree to a weighted tree ensemble, introduces softsign as a differentiable split function, and proposes a novel instance-wise weighting technique. The paper evaluates GRANDE on a predefined benchmark of 19 binary classification datasets and shows that it outperforms existing gradient-boosting and deep learning frameworks on most datasets The paper also demonstrates that GRANDE can learn simple and complex rules within a single model and provide local explanations based on instance-wise weights.

**Strengths:**

1. The transition from GradTree to a Tree ensemble is a natural and straightforward progression, and the reported performance is commendable.

2. The choice of an approximation function for the Heaviside step function is well-justified. The concept of instance-wise weighting is innovative and logically sound.

**Weaknesses:**

My main concern is about the performance evaluation, please find the specific info in the Questions part.

**Questions:**

1. Instead of using exclusively binary class benchmark datasets, could the authors also present comparative results for multi-label classification problems?

2. It would be valuable if the authors could include comparisons with other tree ensemble methods, such as random forest and extra trees.

3. Regarding the interpretability advantages of instance-wise weighting, could the authors provide a more comprehensive analysis? For instance, could they share statistics like the average node count of the highest-weighted estimators across all datasets? Furthermore, could the authors offer any theoretical insights into the benefits of instance-wise weighting?

4. Instance-wise weighting introduces a larger number of weights compared to estimator-wise weighting. Particularly when trees have greater depth (e.g., depth=10, 2^10 vs. 1 for one estimator), is there a significant impact on the training time due to the increased number of weights?

---

> ### Author Response · Authors · 2023-11-23
> **Rebuttal (1 / 4)**
>
> We thank the reviewer for their time and for providing us with promising suggestions!
>
> > Instead of using exclusively binary class benchmark datasets, could the authors also present comparative results for multi-label classification problems?
>
> **TL;DR: GRANDE outperforms existing methods on several multi-class datasets**
>
> In our paper, we focused on binary classification tasks to present a comprehensive evaluation within the limited scope. Yet, we acknowledge the importance of demonstrating that our method is applicable beyond binary classification. Therefore, we conducted additional experiments on several openml multi-class tasks (we used the multi-class datasets from `[1]`) detailed below. We report the results both without HPO and with HPO (30 trials for all methods, constrained by the limited time available during the rebuttal period).
>
> Similar to binary classification, GRANDE outperforms SOTA methods on several datasets for multi-class tasks without additional adjustments of our method. Accordingly, the results for multi-class classification are in line with the results on binary classification tasks presented in the paper. We also included the results in the appendix of our revised version.
>
> HPO (30 trials):
> |  | GRANDE | XGBoost | CatBoost | NODE |
> | -------- | -------- | -------- | -------- | -------- |
> | GesturePhaseSegmentationProcessed      |    0.455 | **0.643** |      0.638 |  0.247 |
> | artificial-characters                  |    0.726 | **0.905** |      0.867 |  0.614 |
> | audiology                              |    0.848 | 0.85  |      0.861 |  **0.89**  |
> | balance-scale                          |    **0.78**  | 0.681 |      0.776 |  0.616 |
> | cnae-9                                 |    0.94  | 0.919 |      0.939 |  **0.942** |
> | jungle_chess_2pcs_raw_endgame_complete |    **0.842** | 0.832 |      0.827 |  0.807 |
> | mfeat-fourier                          |    **0.859** | 0.831 |      0.847 |  0.859 |
> | mfeat-zernike                          |    **0.823** | 0.778 |      0.783 |  0.811 |
> | one-hundred-plants-texture             |    0.818 | 0.741 |      **0.852** |  0.753 |
> | splice                                 |    **0.964** | 0.957 |      0.947 |  0.96  |
> | vehicle                                |    0.817 | 0.809 |      **0.819** |  0.816 |
> | **Normalized Mean**                             |    **0.751** | 0.338 |      0.661 |  0.484 |
> | **Mean Reciprocal Rank (MRR)**                                         |    **0.674** | 0.432 |      0.508 |  0.47  |
> | **Count**                                       |    **5**     | 2     |      2     |  2     |
>
>
> Default Parameters (summarized):
>
> |  | GRANDE | XGBoost  | CatBoost | NODE |
> | -------- | -------- | -------- | -------- | -------- |
> | Normalized Mean | **0.653**     | 0.324    | 0.523         | 0.425     |
> | Mean Reciprocal Rank (MRR)     | **0.674**     | 0.379    | 0.508         | 0.523     |
> | Wins |    **5**     | 1     | 2     | 3     |
>
>
> > It would be valuable if the authors could include comparisons with other tree ensemble methods, such as random forest and extra trees.
>
> We included random forests and extra trees to our evaluation. The results are presented below. Due to lack of space, we were not able to adjust the results table in the main part of our paper, but included them to the Appendix C (Table 13-15). The results are consistent with the claims in our paper: GRANDE achieves superior results both with default parameters and after HPO.
>
> HPO (250 trials):
> |  | GRANDE | XGBoost  | CatBoost | NODE | ExtraTree | RandomForest |
> | -------- | -------- | -------- | -------- | -------- | -------- | -------- |
> | Normalized Mean |    **0.817**     | 0.518     | 0.705     | 0.382     | 0.385     | 0.404     |
> | Mean Reciprocal Rank (MRR)     | **0.668**     | 0.365     | 0.541     | 0.352     | 0.251     | 0.275     |
> | Wins |    **9**     | 2     | 5     | 2     | 0     | 1     |
>
> Default:
> |  | GRANDE | XGBoost  | CatBoost | NODE | ExtraTree | RandomForest |
> | -------- | -------- | -------- | -------- | -------- | -------- | -------- |
> | Normalized Mean                |    **0.652**     | 0.607     | 0.587    | 0.352     | 0.380     | 0.471     |
> | Mean Reciprocal Rank (MRR)     | **0.595**     | 0.450     | 0.416     | 0.339     | 0.283     | 0.371     |
> | Wins                           |    **9**     | 3     | 2     | 2     | 0     | 3     |

---

> ### Author Response · Authors · 2023-11-23
> **Rebuttal (2 / 4)**
>
> > Regarding the interpretability advantages of instance-wise weighting, could the authors provide a more comprehensive analysis? For instance, could they share statistics like the average node count of the highest-weighted estimators across all datasets?
>
> **TL;DR: We provided additional statistics (node count, estimator frequency, skewness, kurtosis, etc.) supporting the claims from our paper.**
>
> [Also see response to `JdQZ`]
>
> Within our paper, we showed in an ablation study that our instance-wise weighting has a positive impact on the performance of GRANDE. In addition, we showcased a real-world scenario where the weighting significantly increases local interpretability by enabling the model to learn representations for simple and complex rules. We agree with the reviewer that a more comprehensive analysis of the weighting would be beneficial and therefore provide additional statistics. We summarize the main results below and provide detailed statistics for each dataset, along with an additional discussion in Appendix D.
>
> The following statistics were obtained by calculating the post-softmax weights for each sample and averaging the values over all samples. In addition, we calculated the same statistics for the top 5% of samples with the highest weights. This provides additional insights as we argue that unique local interactions might only exist for a subset of the samples.
>
>
> | statistic | average value|
> | -------- | -------- |
> | **average internal node count of highest weighted estimator**     | 2.918 |
> | **average leaf node count of highest weighted estimator**     | 3.918  |
> | **fraction of samples for highest weighted estimator**    | 0.325  |
> | **fraction of samples for highest weighted estimator top 5% samples**     | 0.631  |
> | **number of highest weighted estimators**    | 44.221  |
> | **number of highest weighted estimators top 5% samples**    | 5.347  |
>
> We can observe that on average, the highest weighted estimator comprises a small number of ~3 internal and ~4 leaf nodes, allowing for easy interpretation. Furthermore, on average, the highest weighted estimator is the same for ~33% of the data instances with a total of ~44 different estimators having the highest weight for at least one instance. If we further inspect the top 5% of instances with the highest weight for a single estimator, we can observe that the same estimator has the highest weight for ~63% of these instances and only ~5 different estimators achieve the highest weight for at least one instance. This indicates the presence of a small number of local experts for most datasets, having high weights for certain instances where the local rules can be applied.
>
>
> In the subsequent analysis, we evaluate the distribution of the weights in more detail, focusing on skewness as a measure of distribution asymmetry and kurtosis as a measure of tailedness. In general, it stands out that both values increase significantly when considering only the top 5% of samples with the highest weight for a single estimator. This again indicates the presence of local expert estimators for a subset of the data, where unique local interactions were identified. As this is very dataset-specific, we will give a more detailed analysis in the following. The exact values for each dataset are summarized in the Appendix D.
>
>
> | statistic | average value|
> | -------- | -------- |
> | **skewness (absolute)**     | 0.933  |
> | **skewness top 5% (absolute)**     | 1.400  |
> | **(excess) kurtosis**     | 1.291  |
> | **(excess) kurtosis top 5%**     | 3.630  |

---

> > ### Author Response · Authors · 2023-11-23
> > **Rebuttal (3 / 4)**
> >
> > 1. **Skewness:**
> >
> >
> >     In total, for 16/19 datasets the weights exhibit either a skewed (10) or very skewed (6) distribution, i.e. are long-tailed **(a)**. In general, right-skewed distributions are more desired as they indicate a small number of trees having high weights (which we consider as "local experts"), while the majority of estimators have small weights. Considering the top 5% of samples with the highest weights of a single estimator, we can see that the number of very skewed distributions increases from 6 to 13, indicating samples that can be assigned to a class more easily based on a small number of trees (similar to the example in the case study).
> >
> >
> >     | skewness | very skewed; (-∞, -1.0) or (1.0, ∞) | skewed; [-1.0, -0.5) or (0.5, 1.0] | slight skew; [-0.5, 0.5] | sum
> >     | -------- | -------- | -------- | -------- | -------- |
> >     | **left skew**     | 0     | 1     | 1     | **2**     |
> >     | **right skew**     | 6     | 9     | 2     | **17**     |
> >     | **sum**     | **6**     | **10**     | **3**     |      |
> >
> >     | skewness top 5% | very skewed; (-∞, -1.0) or (1.0, ∞) | skewed; [-1.0, -0.5) or (0.5, 1.0] | slight skew; [-0.5, 0.5] | sum
> >     | -------- | -------- | -------- | -------- | -------- |
> >     | **left skew**     | 0     | 1     | 1     | **2**     |
> >     | **right skew**     | 13     | 1     | 3     | **17**     |
> >     | **sum**     | **13**     | **2**     | **4**     |      |
> >
> > 2. **(Excess) Kurtosis:**
> >
> >
> >     We can observe that 11/19 distributions are leptokurtic, i.e. have heavy tails, with 4 considered as very leptokurtic. In general, distributions with heavy tails are interesting, as this indicates outliers (estimators with significantly higher / lower weights). Trees comprising significantly higher weights can be considered as local experts that have learned unique local interactions. Again when considering the top 5% of samples with the highest weights of a single estimator, the number of leptokurtic distributions increases from 11 to 15 with the number of very leptokurtic distributions increasing from 4 to 8. Again, this indicates that there exists samples that can be assigned to a class more easily with simple rules (from "local experts") in many datasets.
> >
> >     | excess kurtosis | count | count top 5% |
> >     | -------- | -------- | -------- |
> >     | **very leptokurtic (3.0, ∞)**    | 4     | 8 |
> >     | **moderatly leptokurtic (0.5, 3.0]**    |   7   | 6 |
> >     | **mesokurtic [-0.5, 0.5]**    | 5     | 3 |
> >     | **platykurtic (-∞, -0.5)**   |   3   | 2 |
> >
> >
> > Overall, the additional statistics are in line with the discussion in the paper. For several datasets, the distribution of weight is long- and/or heavy-tailed indicating the presence of "local expert" estimators. However, the advantage of learning local expert estimators is dataset-dependent and not feasible in all scenarios. This is also confirmed by our results as there are datasets where the distribution is symmetric and mesokurtic. This is valid for instance for numerai28.6, which is a very complex dataset without many simple rules (as indicated by the poor performance of all methods).
> >
> > This additional analysis has been included in Appendix D. However, it should be noted that a more in-depth analysis is necessary to confirm these findings, which we consider beyond the scope of this paper.

---

> > > ### Author Response · Authors · 2023-11-23
> > > **Rebuttal (4 / 4)**
> > >
> > > > Furthermore, could the authors offer any theoretical insights into the benefits of instance-wise weighting?
> > >
> > > Our instance-wise weighting is motivated by the concept of learning local expert models, detailed in the following:
> > >
> > > 1. By assigning weights to each leaf, GRANDE allows each tree to specialize in different parts of the data space. This means trees can focus on areas where they make the most accurate predictions, leading to a more flexible and tailored ensemble.
> > > 2. This weighting scheme promotes diversity within the ensemble, as different trees become experts in different local interactions within the dataset, in contrast to traditional ensembles where each tree contributes equally to the final decision for all instances.
> > > 3.  In terms of managing complexity, GRANDE efficiently learns compact representations for simpler rules, as opposed to more complex models that often learn overly complex representations for these rules.
> > >
> > > > Instance-wise weighting introduces a larger number of weights compared to estimator-wise weighting. Particularly when trees have greater depth (e.g., depth=10, 2^10 vs. 1 for one estimator), is there a significant impact on the training time due to the increased number of weights?
> > >
> > > **TL;DR: There is no significant impact of the instance-wise weighting on the runtime.**
> > >
> > > We understand the concerns of the impact on the runtime. Fortunately, the weighting process is highly efficient and has only a minor impact on the runtime. This is mainly caused by the fact that $\mathbb{L}$ is already calculated to select the probability of the corresponding leaf and therefore only one additional matrix multiplication $\mathbb{R}^{E \times 2^d} \times \mathbb{R}^{E \times 2^d}$ is necessary. A comparison of the runtimes can be found in the following:
> > >
> > > |                      | Instance-Wise Weighting | Estimator Weights |
> > > | --------             | -------- | -------- |
> > > | Mean Runtime (s)        | 46.51    | 35.44    |
> > > | Mean time per epoch (s)  | 0.84     | 0.81     |
> > >
> > > In this case, the runtime per epoch is the relevant measure, as the increase in overall runtime is attributed not only to the additional weights but also to a longer optimization process, which results in increased performance, as shown in Table 11.
> > >
> > >
> > > `[1]` McElfresh, Duncan C., et al. ‘When Do Neural Nets Outperform Boosted Trees on Tabular Data?’ Thirty-Seventh Conference on Neural Information Processing Systems Datasets and Benchmarks Track, 2023, https://openreview.net/forum?id=CjVdXey4zT.

---

### Official Review · Reviewer_Tbch · 2023-10-26

**Soundness:** 3 good
**Presentation:** 3 good
**Contribution:** 3 good
**Rating:** 8
**Confidence:** 4

**Summary:**

This article present "GRANDE" a gradient-based extension of GradTree to weighed ensembles of trees.
"GradTree: Learning Axis-Aligned Decision Trees with Gradient Descent" is a (yet-unpuplished to the best of my knowledge) work by (Marton et al. 2023) that proposes to learn by gradient descent on a relaxation of decision trees.
This relaxation works by reformulating the decision tree as a sum of indicative functions (one for each leaf), and relaxing the decision thresholds with a "straight-through" (ST) operator that replaces the hard decision by a soft decision during the backward pass of gradient descent.
The article first summarizes GradTree, then introduces the specificities of GRANDE:
- a novel variant soft-thresholding called "softsign" that is used with the straigh-through operator
- a novel instance-wise weighing scheme for ensemble of trees

Then a thorough experimental study is provided to evaluate the model and other models on several datasets. The appendix provides several experiments and ablation studies.

**Strengths:**

Dealing with tabular data, as efficiently as gradient-boosted trees do, though neural networks and gradient descent is yet an open challenge. For this very reason, proposing new, or even slightly new models that are able to train tree ensembles in a reasonable time through gradient descent is an interesting contribution.

- The paper is clear, well written, and illustrated with several illustrating Figures. I liked reading it.
- I could not manage to run the supplementary material code, but the provided experiments seems serious and convincing with several datasets and models tested and some ablation studies
- This method, although very slow at training time when compared to gradient-boosting, seems promising for the many neural/tree hybridizations its suggests
- The instance-based weighing is interesting (especially for explainability)

**Weaknesses:**

- My major concern is about hyper-parameters tuning (section C of appendix): I understand that compute resources should be spared, but it seems unfair to optimize the number of trees for GRANDE but not for XGBoost and CatBoost, especially given the fact that XGBoost and CatBoost are the cheapest algorithms to train.
- The results of GRANDE are good on several datasets, but become less impressive when the number of features is high
- The 2^d term in the sums suggests that the depth is a real limitation of the method
- Fact is that gradient descent is much slower than greedy optimization
- The related-works part seems a bit short given the huge literature on ensemble trees

Minor Remarks:
page 4 (section 3.3) : the W matrix is in R^{E\times 2^d}, not R^E \times R^{2^d}
Regarding the code I tried unsuccessfully to run the notebooks: two packages were missing in the installation script: "chardet" and "cchardet", even with this that fixed there is a remaining issue with the "TabSurvey" sub-package which seems to require a specific environment to run (It fails on "from models import" statements).
page 7:  the Phishing Website study should appear as a subsection

**Questions:**

- Your weighing scheme seems interesting but does it really require gradient descent ? Could you take a fixed forest, add weights to the leaves and optimize these weights afterwards ? I am thinking of a paper like (Cui et al. 2023) https://arxiv.org/pdf/2304.13761.pdf
- Did you consider hybridizing greedy and gradient approaches ?

---

> ### Author Response · Authors · 2023-11-23
> **Rebuttal (1 / 2)**
>
> We appreciate the reviewer's constructive feedback and further thank the reviewer for highlighting the strengths of the proposed method!
>
> > My major concern is about hyper-parameters tuning (section C of appendix): I understand that compute resources should be spared, but it seems unfair to optimize the number of trees for GRANDE but not for XGBoost and CatBoost, especially given the fact that XGBoost and CatBoost are the cheapest algorithms to train.
>
> **TL;DR: Tuning the number of estimators for XGBoost and CatBoost has only minor impact on the performance and the results remain unchanged.**
>
> We understand the concerns. The decision to use early stopping instead of tuning the number of estimators was made in line with related work `[1]` to increase search efficiency (not only based on the runtime). Yet, we conducted an additional experiment where we also tuned the number of estimators. Overall, this did not significantly impact the performance of tree-based methods: The average performance slightly decreased when tuning the estimators for both XGBoost (by 0.0010) and CatBoost (by 0.0003). Consequently, for both methods, performance slightly increased in 10 out of 19 datasets and decreased in the remaining 9. In the following, we summarized the comparison of all methods when tuning the number of estimators. The complete results can be found in the appendix of the revised paper (Table 21 and Table 22). These additional results are consistent with the claims in our paper.
>
>
>
> |  | GRANDE | XGBoost | CatBoost | NODE |
> | -------- | -------- | -------- |-------- | -------- |
> | **Mean Reciprocal Rank (MRR) (new, with tuning estimators)**     | **0.684**     | 0.434     | 0.566     | 0.399     |
> | **Mean Reciprocal Rank (MRR) (old, without tuning estimators)**     | **0.702**     | 0.417     | 0.570     | 0.395     |
> | **Normalized Mean (new, with tuning estimators)**     | **0.773**     | 0.436     | 0.629     | 0.285     |
> | **Normalized Mean (old, without tuning estimators)**      | **0.776**     | 0.483     | 0.671     | 0.327     |
> | **Wins (new, with tuning estimators)** |    **9**     | 2     | 6     | 2     |
> | **Wins (old, without tuning estimators)** |    **9**     | 4     | 4     | 2     |
>
>
> > The results of GRANDE are good on several datasets, but become less impressive when the number of features is high
>
> In general, our method excels in handling datasets with a small to medium number of features. For very high-dimensional datasets, learning GRANDE becomes more challenging (as the matrices $I \in \mathbb{R}^{2^d-1 \times n}$ and $T \in \mathbb{R}^{2^d-1 \times n}$ scale with the number of features). This challenge is addressed by using a feature subset for each estimator, supported by our instance-wise weighting, which allows assigning higher weights to estimators focusing on more important features. Although GRANDE showed comparatively modest results on the *madelon* dataset (500 features) and the *Bioresponse* dataset (1,776 features), it outperformed other methods on the SpeedDating dataset (256 features).
>
> > The 2^d term in the sums suggests that the depth is a real limitation of the method.
>
> Theoretically, depth is a limitation of the method. However, in practice, GRANDE can be efficiently trained up to a depth of 10. Preliminary experiments indicated that a depth of 6 is sufficient to achieve SOTA performance, as increasing the depth further does not significantly impact performance.

---

> > ### Author Response · Authors · 2023-11-23
> > **Rebuttal (2 / 2)**
> >
> > > Fact is that gradient descent is much slower than greedy optimization
> >
> > We agree that a gradient-based optimization is slower than a greedy optimization, as we discussed in Section 4.2. However, using gradient-based optimization with GRANDE significantly enhances performance on several datasets while typically maintaining a reasonable runtime of less than 60 seconds for most datasets. Furthermore, gradient-based optimization offers additional flexibility (integration of custom loss functions and compatibility with deep learning architectures, for example, multimodal learning) compared to greedy methods.
> >
> > > The related-works part seems a bit short given the huge literature on ensemble trees
> >
> > We acknowledge that due to the limited scope, the related work section is comparatively brief. In the revised version of our paper, we have included additional references to gradient-based trees `[2,3]`, greedy tree ensembles `[4,5]` and tree weighting schemes `[6,7]`. Furthermore, we conducted additional experiments including further greedy tree ensemble methods (RandomForest and ExtraTree) `[SMN1]` and SAINT as new DL benchmark `[Duhp]`. For a more detailed discussion of existing gradient-based tree ensemble methods, please refer to our response to reviewer `Duhp`.
> >
> > > Minor Remarks: page 4 (section 3.3) : the W matrix is in R^{E\times 2^d}, not R^E \times R^{2^d}
> >
> > Thank you for this notification. We have made the necessary adjustments in the revised version of our paper.
> >
> > > Regarding the code I tried unsuccessfully to run the notebooks: two packages were missing in the installation script: "chardet" and "cchardet", even with this that fixed there is a remaining issue with the "TabSurvey" sub-package which seems to require a specific environment to run (It fails on "from models import" statements).
> >
> > Unfortunately, we were not able to reproduce the error. The error seems to come from the TabSurvey package, which we used to access the implementation of NODE. However, it should be possible to run the experiments without NODE by omitting the corresponding import statement and setting the variable in our configuration to false (`config['benchmarks']['NODE'] = false`).
> >
> >
> > > page 7: the Phishing Website study should appear as a subsection
> >
> > We adjusted this in the revised paper version.
> >
> >
> > > Your weighing scheme seems interesting but does it really require gradient descent ? Could you take a fixed forest, add weights to the leaves and optimize these weights afterwards ? I am thinking of a paper like (Cui et al. 2023) https://arxiv.org/pdf/2304.13761.pdf
> >
> > Taking a fixed forest and then optimizing the leaf weights, akin to the approach of Cui et al. (2023), is indeed a viable strategy and can be viewed as a variant of the method described in our paper. However, this would separate the process of constructing the trees from learning the weighting scheme.
> >
> > In GRANDE's method, gradient descent is used to optimize the entire model structure, including the tree construction and the weights, in an end-to-end manner. This contrasts with using a predefined forest and optimizing only the leaf weights.
> >
> > The key difference here is that in GRANDE, the tree structure and the weights are optimized together, therefore allowing individual trees to learn unique local interactions (as highlighted in the case study) during the training. This is only possible if the weighting is incorporated in the training procedure and not when learning the weights post-hoc. We included a short discussion in the section on out instance-wise weighting in the revised version of our paper.
> >
> >
> > > Did you consider hybridizing greedy and gradient approaches?
> >
> > In this paper, we did not explore hybridizing greedy and gradient-based approaches. However, this presents an interesting research direction for future work.
> >
> >
> >
> > `[1]` Borisov, Vadim, et al. "Deep neural networks and tabular data: A survey." IEEE Transactions on Neural Networks and Learning Systems (2022).
> >
> > `[2]` Yang, Yongxin, Irene Garcia Morillo, and Timothy M. Hospedales. "Deep neural decision trees." arXiv preprint arXiv:1806.06988 (2018).
> >
> > `[3]` Hazimeh, Hussein, et al. "The tree ensemble layer: Differentiability meets conditional computation." International Conference on Machine Learning. PMLR, 2020.
> >
> > `[4]` Geurts, Pierre, Damien Ernst, and Louis Wehenkel. "Extremely randomized trees." Machine learning 63 (2006): 3-42.
> >
> > `[5]` Breiman, Leo. "Random forests." Machine learning 45 (2001): 5-32.
> >
> > `[6]` Yu, Gyeong-In, et al. "WindTunnel: towards differentiable ML pipelines beyond a single model." Proceedings of the VLDB Endowment 15.1 (2021): 11-20.
> >
> > `[7]` He, Xinran, et al. "Practical lessons from predicting clicks on ads at facebook." Proceedings of the eighth international workshop on data mining for online advertising. 2014.

---

### Official Review · Reviewer_JdQZ · 2023-10-31

**Soundness:** 3 good
**Presentation:** 3 good
**Contribution:** 3 good
**Rating:** 6
**Confidence:** 4

**Summary:**

This paper proposes a novel approach for learning hard, axis-aligned decision tree ensembles using end-to-end gradient descent based on the straight-through operator. The core contributions are two folds: 1) using an alternative differentiable split function (softsign); 2) introducing an advanced instance-wise weighting mechanism for tree ensemble. Experiments on tabular benchmark show that this method outperforms existing gradient-boosting and deep learning frameworks.

**Strengths:**

1. This is one of the few deep learning based works which beat XGB on tabular benchmark.
2. The contributions (alternative differentiable split function and instance-wise weighting) are supported by ablation experiments.
3. It provides all the hyperparameters in appendix, which helps reproduction.

**Weaknesses:**

1. This paper lacks further analysis for instance-wise weighting. Because the final results are weighted by Softmax, the prediction of each tree is not separate now. If we cut off one tree, the contributions of the other tree are also changed. This is different from XGB and NODE, but the authors did not point out it. Moreover, it is better to analysis the distribution of instance weights. For example:

a) Is it long-tailed?

b) Are some trees very important for most of the samples?

2. Too many hyperparameters, i.e. 11 are tuned for the proposed method. While the number of hyperparameters is no more than 4 for compared methods. We have reason to doubt that the performance gain is obtained by finer hyperparameter tuning. Could you reduce your number of hyperparameters to 4 and retrain your model for comparison?

**Questions:**

See "Weaknesses".

---

> ### Author Response · Authors · 2023-11-23
> **Rebuttal (1 / 3)**
>
> We thank the reviewer for their time and providing us with valuable feedback!
>
> > Because the final results are weighted by Softmax, the prediction of each tree is not separate now. If we cut off one tree, the contributions of the other tree are also changed. This is different from XGB and NODE, but the authors did not point out it.
>
> Thank you for pointing this out. Indeed, when employing a softmax function to calculate the weights for GRANDE, the individual tree predictions are not separated. Therefore, the contributions of individual trees change. This occurs not only in scenarios such as removing one tree from the ensemble but also during instance-wise weighting: Assume we have two data instances that are very similar and the prediction of our ensemble differs only in one tree. For this particular tree, depending on the different paths taken by the samples, we select two distinct logits. Consequently, when calculating the softmax to generate the weights, the contribution of each tree changes.
>
> This aspect is crucial in enabling the learning of "local experts" within GRANDE. If the path capturing a unique local interaction is traversed by a sample, the corresponding tree should have a very high weight, resulting in reduced weights for the remaining trees.
>
> We acknowledge that this aspect was not adequately addressed in the current version of the paper, and have therefore provided clarification in the revised version (Section 3.3).
>
> > Moreover, it is better to analysis the distribution of instance weights.
> For example:
> a) Is it long-tailed?
> b) Are some trees very important for most of the samples?
>
> **TL;DR: (a) In most cases, the distributions are long-tailed (and heavy-tailed). (b) On average, for 38% of the samples, the same tree is the most important one, but this is very dataset- and instance-specific (see discussion below).**
>
> [Also see response to `SMN1`]
>
> Within our paper, we showed in an ablation study that our instance-wise weighting has a positive impact on the performance of GRANDE. In addition, we showcased a real-world scenario where the weighting significantly increases local interpretability by enabling the model to learn representations for simple and complex rules. We agree with the reviewer that a more comprehensive analysis of the weighting would be beneficial and therefore provide additional statistics. We summarize the main results below and provide detailed statistics for each dataset, along with an additional discussion in Appendix D.
>
> The following statistics were obtained by calculating the post-softmax weights for each sample and averaging the values over all samples. In addition, we calculated the same statistics for the top 5% of samples with the highest weights. This provides additional insights as we argue that unique local interactions might only exist for a subset of the samples.
>
>
> | statistic | average value|
> | -------- | -------- |
> | **average internal node count of highest weighted estimator**     | 2.918 |
> | **average leaf node count of highest weighted estimator**     | 3.918  |
> | **fraction of samples for highest weighted estimator**    | 0.325  |
> | **fraction of samples for highest weighted estimator top 5% samples**     | 0.631  |
> | **number of highest weighted estimators**    | 44.221  |
> | **number of highest weighted estimators top 5% samples**    | 5.347  |
>
> We can observe that on average, the highest weighted estimator comprises a small number of ~3 internal and ~4 leaf nodes, allowing for easy interpretation. Furthermore, on average, the highest weighted estimator is the same for ~33% of the data instances with a total of ~44 different estimators having the highest weight for at least one instance. If we further inspect the top 5% of instances with the highest weight for a single estimator, we can observe that the same estimator has the highest weight for ~63% of these instances and only ~5 different estimators achieve the highest weight for at least one instance. This indicates the presence of a small number of local experts for most datasets, having high weights for certain instances where the local rules can be applied.

---

> > ### Author Response · Authors · 2023-11-23
> > **Rebuttal (2 / 3)**
> >
> > In the subsequent analysis, we evaluate the distribution of the weights in more detail, focusing on skewness as a measure of distribution asymmetry and kurtosis as a measure of tailedness. In general, it stands out that both values increase significantly when considering only the top 5% of samples with the highest weight for a single estimator. This again indicates the presence of local expert estimators for a subset of the data, where unique local interactions were identified. As this is very dataset-specific, we will give a more detailed analysis in the following. The exact values for each dataset are summarized in the Appendix D.
> >
> >
> > | statistic | average value|
> > | -------- | -------- |
> > | **skewness (absolute)**     | 0.933  |
> > | **skewness top 5% (absolute)**     | 1.400  |
> > | **(excess) kurtosis**     | 1.291  |
> > | **(excess) kurtosis top 5%**     | 3.630  |
> >
> > 1. **Skewness:**
> >
> >
> >     In total, for 16/19 datasets the weights exhibit either a skewed (10) or very skewed (6) distribution, i.e. are long-tailed **(a)**. In general, right-skewed distributions are more desired as they indicate a small number of trees having high weights (which we consider as "local experts"), while the majority of estimators have small weights. Considering the top 5% of samples with the highest weights of a single estimator, we can see that the number of very skewed distributions increases from 6 to 13, indicating samples that can be assigned to a class more easily based on a small number of trees (similar to the example in the case study).
> >
> >
> >     | skewness | very skewed; (-∞, -1.0) or (1.0, ∞) | skewed; [-1.0, -0.5) or (0.5, 1.0] | slight skew; [-0.5, 0.5] | sum
> >     | -------- | -------- | -------- | -------- | -------- |
> >     | **left skew**     | 0     | 1     | 1     | **2**     |
> >     | **right skew**     | 6     | 9     | 2     | **17**     |
> >     | **sum**     | **6**     | **10**     | **3**     |      |
> >
> >     | skewness top 5% | very skewed; (-∞, -1.0) or (1.0, ∞) | skewed; [-1.0, -0.5) or (0.5, 1.0] | slight skew; [-0.5, 0.5] | sum
> >     | -------- | -------- | -------- | -------- | -------- |
> >     | **left skew**     | 0     | 1     | 1     | **2**     |
> >     | **right skew**     | 13     | 1     | 3     | **17**     |
> >     | **sum**     | **13**     | **2**     | **4**     |      |
> >
> > 2. **(Excess) Kurtosis:**
> >
> >
> >     We can observe that 11/19 distributions are leptokurtic, i.e. have heavy tails, with 4 considered as very leptokurtic. In general, distributions with heavy tails are interesting, as this indicates outliers (estimators with significantly higher / lower weights). Trees comprising significantly higher weights can be considered as local experts that have learned unique local interactions. Again when considering the top 5% of samples with the highest weights of a single estimator, the number of leptokurtic distributions increases from 11 to 15 with the number of very leptokurtic distributions increasing from 4 to 8. Again, this indicates that there exists samples that can be assigned to a class more easily with simple rules (from "local experts") in many datasets.
> >
> >     | excess kurtosis | count | count top 5% |
> >     | -------- | -------- | -------- |
> >     | **very leptokurtic (3.0, ∞)**    | 4     | 8 |
> >     | **moderatly leptokurtic (0.5, 3.0]**    |   7   | 6 |
> >     | **mesokurtic [-0.5, 0.5]**    | 5     | 3 |
> >     | **platykurtic (-∞, -0.5)**   |   3   | 2 |
> >
> >
> > Overall, the additional statistics are in line with the discussion in the paper. For several datasets, the distribution of weight is long- and/or heavy-tailed indicating the presence of "local expert" estimators. However, the advantage of learning local expert estimators is dataset-dependent and not feasible in all scenarios. This is also confirmed by our results as there are datasets where the distribution is symmetric and mesokurtic. This is valid for instance for numerai28.6, which is a very complex dataset without many simple rules (as indicated by the poor performance of all methods).
> >
> > This additional analysis has been included in Appendix D. However, it should be noted that a more in-depth analysis is necessary to confirm these findings, which we consider beyond the scope of this paper.

---

> ### Author Response · Authors · 2023-11-23
> **Rebuttal (3 / 3)**
>
> > Too many hyperparameters, i.e. 11 are tuned for the proposed method. While the number of hyperparameters is no more than 4 for compared methods. We have reason to doubt that the performance gain is obtained by finer hyperparameter tuning. Could you reduce your number of hyperparameters to 4 and retrain your model for comparison?
>
> **TL,DR: When reducing the number of hyperparameters or the number of trials, GRANDE still achieves SOTA performance**
>
> We understand the concerns regarding the number of hyperparameters. We already tried to account for this with the default hyperparameter experiment included in the original version of the paper (Table 3 and Table 8), demonstrating that GRANDE achieved superior results even without hyperparameter optimization. Acknowledging the importance of this topic, we have included two additional experiments focusing on hyperparameter optimization:
> 1. **Reduced number of trials to 30:** We reduced the number of trials in the HPO from originally 250 trials to 30 for all methods (similar to `[1]`) to compare the performance with a less extensive search. We can observe that GRANDE has the greatest benefit when increasing the number of trials. Yet, even with only 30 trials, GRANDE achieves the highest mean reciprocal rank (MRR) and normalized mean, as well as the most wins. The detailed results on all datasets along with a discussion are included in the appendix of the revised paper version and summarized in the following table:
>
>     |  | GRANDE | XGBoost  | CatBoost | NODE |
>     | -------- | -------- | -------- | -------- | -------- |
>     | Normalized Mean                           |    **0.694** | 0.430 |      0.676  |  0.336 |
>     | Mean Reciprocal Rank (MRR)                                       |    **0.636**  | 0.434 |      0.579 |  0.434 |
>     | Count                                     |    **8**     | 3     |      5     |  3     |
>
>
> 2. **Reduce grid for GRANDE to 4 parameters:** Following the reviewer's suggestion, we reduced the number of hyperparameters for GRANDE to four (one learning_rate, dropout, loss, and cosine_decay) and conducted an additional HPO with 250 trials. Unfortunately, we could not complete the search for all datasets during the rebuttal period (*nomao* and *adult* datasets are pending and will be included in the camera-ready version). Reducing the grid results in a slightly worse performance for GRANDE. Especially, optimizing separate learning rates for different components (split values, split indices, leaf probabilities, and leaf weights) using a larger grid did further enhances GRANDE's performance. Yet, the results are in line with the claims from the paper and GRANDE still has the highest normalized mean, MRR and number of wins.
>
>
>     |  | GRANDE | XGBoost | CatBoost | NODE |
>     | -------- | -------- | -------- | -------- | -------- |
>     | Normalized Mean                           |    **0.736** | 0.428 |      0.710 |  0.364 |
>     | Mean Reciprocal Rank (MRR)                                       |    **0.652** | 0.358 |      0.608 |  0.466 |
>     | Count                                     |    **7**     | 0     |      6     |  4     |
>
>
> `[1]` McElfresh, Duncan C., et al. ‘When Do Neural Nets Outperform Boosted Trees on Tabular Data?’ Thirty-Seventh Conference on Neural Information Processing Systems Datasets and Benchmarks Track, 2023, https://openreview.net/forum?id=CjVdXey4zT.

---

### Official Review · Reviewer_Duhp · 2023-11-02

**Soundness:** 3 good
**Presentation:** 3 good
**Contribution:** 2 fair
**Rating:** 6
**Confidence:** 3

**Summary:**

The paper extends the GradTree support of Marton to weighted ensembles of trees, and using a different splitting function based on the soft-sign.
The paper compares their results to XGBoost, CatBoost and NODE, and show that they outperform these based on average rank of F1 score.

**Strengths:**

The paper gives a detailed and clear description of the approach. The experimental evaluation and the evaluation protocol are well-defined and sound, and the results look promising.

**Weaknesses:**

Given the popularity of gradient-based tree models in recent years, I feel like a more thorough comparison with competing methods would be warranted. In particular relating this work to work on learning weighting for fixed tree structures would be interesting, as first discussed in "Practical Lessons from Predicting Clicks on Ads at Facebook" by He et.al.
"Deep Neural Decision Trees" by Yang et al also seems relevant, as well as "Deep Neural Decision Forests" by Kontschieder et al, "SDTR: Soft Decision Tree Regressor for Tabular Data" by Luo and ". The tree ensemble layer: Differentiability meets conditional computation." by Hazimeh et al.


"WindTunnel: Towards Differentiable ML Pipelines Beyond a Single Model" also seems closely related, though they only fine-tune existing tree models with gradient descent.

I find it also somewhat confusing that it is claimed that gradient boosted models are the defacto state-of-the-art on tabular data, when there has been a lot of recent work on neural networks for tabular data, often outperforming gradient boosting models, see "Well-tuned Simple Nets Excel on Tabular Datasets" by Kadra for example. It would be great to include at least one other neural approach in the comparison.


For the evaluation, I would have expected performance profiles or critical difference diagrams based on AUC or AP, instead of the mean and rank (or in addition to it, though the unnormalized mean is not a good way to aggregate performance across datasets).

Overall, I find the contribution beyond Borisov somewhat slim, though that work is not published at a conference. If these are the same authors, I would recommend combining the two into a single submission.

Minor:
The cylinder-bands dataset is called out, it would be interesting to see if this is the version with target leakage via the job_number column, or without that column.
The "straight-through operator" is simply the subgradient of the max, and I think the standard reference for that is "Evaluation of pooling operations in convolutional architectures for object recognition" by Scherer et.al.

**Questions:**

How do the results in Table 4 compare to using the hard split function?

---

> ### Author Response · Authors · 2023-11-23
> **Rebuttal (Part 1 / 3)**
>
> We thank the reviewer for their time and valuable feedback!
>
> > Given the popularity of gradient-based tree models in recent years, I feel like a more thorough comparison with competing methods would be warranted. In particular relating this work to work on learning weighting for fixed tree structures would be interesting, as first discussed in "Practical Lessons from Predicting Clicks on Ads at Facebook" by He et.al. "Deep Neural Decision Trees" by Yang et al also seems relevant, as well as "Deep Neural Decision Forests" by Kontschieder et al, "SDTR: Soft Decision Tree Regressor for Tabular Data" by Luo and ". The tree ensemble layer: Differentiability meets conditional computation." by Hazimeh et al.
>
> We are aware that due to lack of space, the related work section was comparatively short and did not cover all related, gradient-based tree methods in detail. We adjusted the related work section (and the section on instance-wise weighting) to cover the mentioned works and provide a more detailed differentiation of our approach as follows:
> * **"Practical Lessons from Predicting Clicks on Ads at Facebook"**: They use GBDTs as feature transformers to generate categorical input features for a sparse linear classifier which can be seen as learning a weighting for the GBDT. There are two main differences to our work (now discussed in Section 3.3):
>     1. Instead of learning a post-hoc weighting, we incorporate the weighting into the training procedure.
>     2. We learn instance-wise weights that are calculated individually for each instance based on the selected paths in the ensemble.
>
>     These two aspects allow learning unique local interactions ("local experts") during the training of our model as showcased in the case study (Section 4.3).
>
> * **"Deep Neural Decision Trees"**: DNDTs in contrast to GRANDE uses soft trees. However, similar to our trees, they use axis-aligned splits which differentiates DNDTs to most hierarchical, gradient-based methods. The GradTree paper `[6]` also discussed the similarities and differences and includes an empirical comparison. During the empirical comparison, they showed that GradTree (the tree structure we adopted) significantly outperformed DNDTs. In the following we will shortly summarize the differences; a more detailed discussion can also be found in `[6]`:
>     1. As already mentioned, the trees are soft.
>     2. The use of the Kronecker product for building the tree with comes with two major limitations:
>         * The tree size depends on the number of features and therefore scales poorly for large datasets (only a maximum number of 12 features per tree is possible according to the authors).
>         * The use of the Kronecker product prevents splitting on the same feature multiple times within one tree which is often important.
>
>
> The following methods have in common that they learn soft, oblique trees. We will first give a short summary of the method and then discuss their differences to GRANDE:
>
> * **"Deep Neural Decision Forests"**: NDFs use a stochastic routing (soft splits) with oblique splits with the goal of guiding representation learning in the lower layers of a CNN.
> * **"SDTR: Soft Decision Tree Regressor for Tabular Data"**: SDTR tries to imitate binary decision trees by a neural model and therefore optimizes soft decision trees (SDTs) e.g. using additional regularization. Their model outperforms non-interpretable MLP structures, but only achieved comparable (slightly worse) performance to non-gradient-based methods (i.e., CatBoost and XGBoost outperformed SDTR).
> * **"The tree ensemble layer: Differentiability meets conditional computation."**: Similarly, they use a stochastic routing and oblique splits with the goal of designing a differentiable layer that can be incorporated in a DL framework.
>
> All the previously mentioned works have a hierarchical tree ensemble structure, achieving differentiability by employing soft and oblique split functions. Therefore, they are very similar to NODE which we benchmarked against in our evaluation as it is according to recent surveys [2,3] the superior hierarchical tree ensemble method. The main difference in the tree structure between existing work and GRANDE is that we employ the ST-operator to maintain hard, axis-aligned splits which provides a beneficial inductive bias (see discussion in Section 3.2 and Section 5).

---

> ### Author Response · Authors · 2023-11-23
> **Rebuttal (Part 2 / 3)**
>
> >  "WindTunnel: Towards Differentiable ML Pipelines Beyond a Single Model" also seems closely related, though they only fine-tune existing tree models with gradient descent.
>
> WindTunnel tries to bring the advantage of a joint, global optimization inherent in DL frameworks to a complete ML pipeline. For GBDTs in the pipeline, this is achieved by smoothing the non-differentiable operations. Thereby, the beneficial inductive bias of axis-aligned splits, which is maintained using GRANDE, is lost. However, we believe that it is an interesting direction for future research to integrate GRANDE in a differentiable pipeline (as proposed by WindTunnel) as this would maintain the inductive bias of axis-aligned splits while still allowing an optimization of the pipeline with gradient descent.
>
> >  I find it also somewhat confusing that it is claimed that gradient boosted models are the defacto state-of-the-art on tabular data, when there has been a lot of recent work on neural networks for tabular data, often outperforming gradient boosting models, see "Well-tuned Simple Nets Excel on Tabular Datasets" by Kadra for example. It would be great to include at least one other neural approach in the comparison.
>
> **TL;DR: We included SAINT as additional gradient-based benchmark and GRANDE demonstrates superior results.**
>
> We are aware that there also exist papers highlighting the strengths of DL methods on tabular data, like the mentioned paper by Kadra et al. However, it is important to note that their evaluation primarily focused on balanced datasets`[1]`, reporting the accuracy and did not consider advanced encoding techniques for GBDT methods. Furthermore, their experiments included an extensive HPO with up to 4 days of time limit for each dataset `[1]`.
> Yet, several recent papers still claim that, even though the gap is diminishing, GBDT are still considered SOTA `[2,3,4]`. At NeurIPS2023 there is an additional survey paper by McElfresh, Duncan C., et al.`[5]` accepted that confirms superior results of GBDT methods over a total of 176 datasets (even though they highlight that the gap to gradient-based methods is diminishing). Based on your feedback, we included Kadra et al. `[1]` to the related work section of our revised paper.
>
> In summary, we agree that an additional, gradient-based benchmark would be beneficial and therefore added SAINT to our evaluation (which is according to `[2]` the superior gradient-based method). The new results are summarized in the following and are also included in the revised paper version (in the appendix due to lack of space in the main part). We provide the results with default parameters and after HPO. Due to the high computational demands of SAINT, we limited the HPO to 30 trials for all methods, which is in line the experiments in `[5]`. Furthermore, SAINT resulted in a OOM error (on an RTX A6000 with 47.55GB VRAM) for *Bioresponse* (1,776 features) which is why we excluded this dataset. The results are summarized in the following and detailed results for each datasets can be found in the Appendix C.
>
> SAINT achieves competitive results on many datasets (best method for one datasets), but also exhibits very poor performance on some datasets (e.g. for *madelon* only the majority class is predicted). As a result, the claims from our paper remain unchanged and GRANDE achieved superior results on most datasets, significantly outperforming existing gradient-based methods.
>
>
> HPO (30 trials):
>
> |  | GRANDE | XGBoost | CatBoost | NODE | SAINT |
> | -------- | -------- | -------- | -------- | -------- | -------- |
> | Normalized Mean                           |    **0.749** | 0.582 |      0.740 |  0.499 |   0.200 |
> | Mean Reciprocal Rank (MRR)                                       |    **0.638** | 0.391 |      0.531 |  0.427 |   0.296 |
> | Count                                     |    **8**     | 2     |      4     |  3     |   1     |
>
> Default Parameters:
>
> |  | GRANDE | XGBoost | CatBoost | NODE | SAINT |
> | -------- | -------- | -------- | -------- | -------- | -------- |
> | Normalized Mean                           |    0.701 | **0.718** |      0.674 |  0.449 |   0.228 |
> | Mean Reciprocal Rank (MRR)                                       |    **0.631** | 0.495 |      0.442 |  0.382 |   0.333 |
> | Count                                     |    **9**     | 3     |      2     |  2     |   2 |

---

> > ### Author Response · Authors · 2023-11-23
> > **Rebuttal (Part 3 / 3)**
> >
> > > For the evaluation, I would have expected performance profiles or critical difference diagrams based on AUC or AP, instead of the mean and rank (or in addition to it, though the unnormalized mean is not a good way to aggregate performance across datasets).
> >
> > Thank you for highlighting this. We agree that the mean is a suboptimal measure and therefore included the normalized mean in addition to the mean reciprocal rank (MRR) to the revised version of our paper. The following table provides a summary of the normalized mean for all tables provided in the paper:
> >
> > |                     | GRANDE   | XGBoost     | CatBoost | NODE.    |
> > | --------            | -------- | -------- | -------- | -------- |
> > | F1 Score (HPO 250 trials)      | **0.776**     | 0.483     | 0.671     | 0.327     |
> > | ROC AUC Score (HPO 250 trials) | **0.769**     | 0.419     | 0.684     | 0.395     |
> > | Accuracy (HPO 250 trials)      | **0.793**     | 0.523     | 0.730     | 0.126     |
> > | F1 Score (Default Parameters)  | **0.637**     | 0.586     | 0.579     | 0.270     |
> >
> > These results, based on the normalized mean, verify GRANDE's strong performance compared to existing methods.
> >
> > In addition, we have added a performance profile to Appendix B of our paper (including images to the response is not possible). The performance profile similarly shows GRANDE's superior performance, consistently ranking above existing methods.
> >
> > > Overall, I find the contribution beyond Borisov somewhat slim, though that work is not published at a conference. If these are the same authors, I would recommend combining the two into a single submission.
> >
> > Based on the comment, it appears that the reviewer is referring to Marton et al. `[6]` rather than Borisov. Therefore, our response will focus on differentiating our work from that of Marton et al. `[6]`.
> >
> > 1. The objective of GradTree and the corresponding paper is on learning interpretable decision trees. We build upon this work to develop a high-performance tree ensemble method, demonstrating through extensive evaluation that GRANDE achieves SOTA results for tabular data.
> > 2. Our work proposes novel, independent and non-trivial contributions, detailed in the following:
> >     * We extend their approach from individual trees to tree ensembles, maintaining efficient computation.
> >     * We worked out a more suitable splitting function for gradient-based optimization of DTs, significantly enhancing GRANDE's performance.
> >     * We propose a novel instance-wise weighting scheme which enables learning unique local interactions within the data and increases the performance and local interpretability.
> >
> > Furthermore, our ablation study demonstrates that these non-trivial extensions are crucial for achieving SOTA performance.
> >
> >
> > > Minor: The cylinder-bands dataset is called out, it would be interesting to see if this is the version with target leakage via the job_number column, or without that column. The "straight-through operator" is simply the subgradient of the max, and I think the standard reference for that is "Evaluation of pooling operations in convolutional architectures for object recognition" by Scherer et.al.
> >
> > **TL;DR: The results when excluding the job_number column are in line with the results presented in the paper**
> >
> > Thank you for highlighting this point, as we were previously unaware of this target leak. We used the cylinder-bands version available on openml which is commonly used in existing benchmarks and indeed includes the "job_number" column. We performed an additional experiment excluding this column with the following results:
> >
> >
> > | | GRANDE | XGBoost | CatBoost | NODE |
> > | -------- | -------- | -------- | -------- | -------- |
> > | with job_number            | **0.819**     | 0.773     | 0.801     | 0.754     |
> > | without job_number         | **0.808**     | 0.762     | 0.787     | 0.746     |
> > | performance difference     | -0.011         | -0.011     | -0.014     | -0.008     |
> >
> >
> > Overall, the exclusion of the "job_number" column led to an approximately equivalent decrease in performance across all methods. Consequently, our claim remains unchanged: GRANDE achieves significantly higher results compared to existing methods for the cylinder-bands dataset.
> >
> >
> > > Questions:
> > How do the results in Table 4 compare to using the hard split function?
> >
> > Sorry for the confusion. The table displays the results of the hard split function obtained by employing the ST operator to discretize the different options for the split function (sigmoid, entmoid or softsign). The choice of the split function therefore only impacts the gradients as discussed in Section 3.2. We have accordingly revised the text in the relevant subsection of our manuscript.

---

> > > ### Author Response · Authors · 2023-11-23
> > > **References**
> > >
> > > `[1]` Kadra, Arlind, et al. "Well-tuned simple nets excel on tabular datasets." Advances in neural information processing systems 34 (2021): 23928-23941.
> > >
> > > `[2]` Borisov, Vadim, et al. "Deep neural networks and tabular data: A survey." IEEE Transactions on Neural Networks and Learning Systems (2022).
> > >
> > > `[3]` Grinsztajn, Léo, Edouard Oyallon, and Gaël Varoquaux. "Why do tree-based models still outperform deep learning on typical tabular data?." Advances in Neural Information Processing Systems 35 (2022): 507-520.
> > >
> > > `[4]` Shwartz-Ziv, Ravid, and Amitai Armon. "Tabular data: Deep learning is not all you need." Information Fusion 81 (2022): 84-90.
> > >
> > > `[5]` McElfresh, Duncan C., et al. ‘When Do Neural Nets Outperform Boosted Trees on Tabular Data?’ Thirty-Seventh Conference on Neural Information Processing Systems Datasets and Benchmarks Track, 2023, https://openreview.net/forum?id=CjVdXey4zT.
> > >
> > > `[6]` Marton, Sascha, et al. ‘GradTree: Learning Axis-Aligned Decision Trees with Gradient Descent’. NeurIPS 2023 Second Table Representation Learning Workshop, 2023, https://openreview.net/forum?id=lWBMNF7D8F.

---

### Author Response · Authors · 2023-11-23
**Rebuttal Summary**

We highly appreciate the reviewers' constructive feedback, and we thank the reviewers for highlighting the strengths and novelty of our method. Specifically, reviewers pointed out the promising empirical performance `[JdQZ, Tbch, SMN1]`, the sound empirical evaluation `[Duhp, JdQZ, Tbch]`, and the clear presentation `[DuhpZ, Tbch]`.

In the following, we will summarize the main points from the discussion with the reviewers. We have also uploaded a revised version of the manuscript incorporating the reviewers' comments.

* **Additional Benchmarks** `[Duhp, SMN1]` **:** We included additional benchmarks based on reviewer feedback: SAINT, RandomForest, ExtraTrees.
    * → GRANDE still outperformed the other methods on most datasets (most wins) and achieved the highest mean reciprocal rank (MRR) and normalized mean (with default parameters as well as HPO).


* **Additional Performance Measurement** `[Duhp]` **:** We added the normalized mean as additional performance measures and a performance profile to substantiate our claims.


* **Hyperparameter Optimization** `[JdQZ, Tbch]` **:** In addition to the extensive HPO with 250 trials including a large grid for GRANDE, we provide the following additional results:
    * **Reduced grid (4 parameters)** → Reducing the grid to 4 parameters for GRANDE resulted in a slightly worse performance compared to using the complete grid for the HPO. Yet, the results are in line with the claims from the paper and GRANDE still has the highest normalized mean, MRR and number of wins.
    * **Reduced number of trials to 30** → When reducing the number of trials to 30 for all methods, we can observe that GRANDE has the highest benefit when increasing the number of trials in the HPO further. Yet, even with only 30 trials, GRANDE achieves the highest MRR and normalized mean, as well as the most wins.
    * **Tune number of estimators for GBDT** → Tuning the estimators for GBDT (instead of setting them to a high value and using early stopping) only has a minor impact on the performance and the results as well as the claims from the paper remain unchanged.


* **Statistics on Weighting** `[JdQZ, SMN1]` **:** We provide additional statistics on our weighting: We verified that distinct weights are learned for most datasets, indicating the presence of local expert estimators in most datasets (as showcased during our case study).


* **Multi-Class Results** `[SMN1]` **:** We show that our approach is easily extensible to further tasks by providing results for multi-class datasets, without needing adjustments to our method. We achieved similar results to binary classification, resulting in a superior performance on most datasets, the highest MRR and normalized mean (with default parameters and HPO).

---

### Meta-Review · Area_Chair_uVGE · 2023-12-09

**Metareview:**

This submission contributes an approach to learn differentiable forest-like architectures with an axis-aligned indcutive bias. It generated much interesting discussion with the reviewers. The reviewers found the work an interesting addition to the literature. They appreciated the sound evidence of performance improvements as well as the clear presentation.

**Justification For Why Not Higher Score:**

The ratings are not that high, and it is not clear to me whether the submission is really revolutionary. I also find that evaluating on only 19 datasets is a limited evaluation. However, I could be convinced to bump this up.

**Justification For Why Not Lower Score:**

It seems to be a solid work, with actual methodological novelty and solid empirical study.

---

### Decision · Program_Chairs · 2024-01-16

Accept (poster)